# Landscape drives zoonotic malaria prevalence in non-human primates

Emilia Johnson[1,2,3]*, Reuben Sunil Kumar Sharma[4], Pablo Ruiz Cuenca[2,5,6], Isabel Byrne[2], Milena Salgado-Lynn[7,8,9], Zarith Suraya Shahar[4], Lee Col Lin[4], Norhadila Zulkifli[4], Nor Dilaila Mohd Saidi[4], Chris Drakeley[10], Jason Matthiopoulos[1], Luca Nelli[1], Kimberly Fornace[1,2,3,11]

[1]School of Biodiversity, One Health and Veterinary Medicine, University of Glasgow, Glasgow, United Kingdom; [2]Department of Disease Control, London School of Hygiene & Tropical Medicine, London, United Kingdom; [3]Centre on Climate Change and Planetary Health, London School of Hygiene & Tropical Medicine, London, United Kingdom; [4]Faculty of Veterinary Medicine, Universiti Putra Malaysia, Selangor, Malaysia; [5]Lancaster University, Bailrigg, Lancaster, United Kingdom; [6]Liverpool School of Tropical Medicine, Pembroke Place Liverpool, Liverpool, United Kingdom; [7]School of Biosciences, Cardiff University, Cardiff, United Kingdom; [8]Wildlife Health, Genetic and Forensic Laboratory, Sabah Wildlife Department, Wisma Muis, Kota Kinabalu, Malaysia; [9]Danau Girang Field Centre, Sabah Wildlife Department, Kinabalu Sabah, Malaysia; [10]Department of Infection Biology, London School of Hygiene & Tropical Medicine, London, United Kingdom; [11]Saw Swee Hock School of Public Health, National University of Singapore, Singapore, Singapore

*For correspondence: emilia.johnson@glasgow.ac.uk

Competing interest: The authors declare that no competing interests exist.

**Abstract** Zoonotic disease dynamics in wildlife hosts are rarely quantified at macroecological scales due to the lack of systematic surveys. Non-human primates (NHPs) host *Plasmodium knowlesi*, a zoonotic malaria of public health concern and the main barrier to malaria elimination in Southeast Asia. Understanding of regional *P. knowlesi* infection dynamics in wildlife is limited. Here, we systematically assemble reports of NHP *P. knowlesi* and investigate geographic determinants of prevalence in reservoir species. Meta-analysis of 6322 NHPs from 148 sites reveals that prevalence is heterogeneous across Southeast Asia, with low overall prevalence and high estimates for Malaysian Borneo. We find that regions exhibiting higher prevalence in NHPs overlap with human infection hotspots. In wildlife and humans, parasite transmission is linked to land conversion and fragmentation. By assembling remote sensing data and fitting statistical models to prevalence at multiple spatial scales, we identify novel relationships between *P. knowlesi* in NHPs and forest fragmentation. This suggests that higher prevalence may be contingent on habitat complexity, which would begin to explain observed geographic variation in parasite burden. These findings address critical gaps in understanding regional *P. knowlesi* epidemiology and indicate that prevalence in simian reservoirs may be a key spatial driver of human spillover risk.

## eLife assessment

This **useful** study presents findings regarding the impact of forest cover and fragmentation on the prevalence of malaria in non-human primates. The evidence supporting the claims of the authors is **solid**.

**eLife digest** Zoonotic diseases are infectious diseases that are transmitted from animals to humans. For example, the malaria-causing parasite *Plasmodium knowlesi* can be transmitted from monkeys to humans through mosquitos that have previously fed on infected monkeys. In Malaysia, progress towards eliminating malaria is being undermined by the rise of human incidences of 'monkey malaria', which has been declared a public health threat by The World Health Organisation.

In humans, cases of monkey malaria are higher in areas of recent deforestation. Changes in habitat may affect how monkeys, insects and humans interact, making it easier for diseases like malaria to pass between them. Deforestation could also change the behaviour of wildlife, which could lead to an increase in infection rates. For example, reduced living space increases contact between monkeys, or it may prevent behaviours that help animals to avoid parasites.

Johnson et al. wanted to investigate how the prevalence of malaria in monkeys varies across Southeast Asia to see whether an increase of *Plasmodium knowlesi* in primates is linked to changes in the landscape. They merged the results of 23 existing studies, including data from 148 sites and 6322 monkeys to see how environmental factors like deforestation influenced the amount of disease in different places.

Many previous studies have assumed that disease prevalence is high across all macaques, monkey species that are considered pests, and in all places. But Johnson et al. found that disease rates vary widely across different regions. Overall disease rates in monkeys are lower than expected (only 12%), but in regions with less forest or more 'fragmented' forest areas, malaria rates are higher. Areas with a high disease rate in monkeys tend to further coincide with infection hotspots for humans. This suggests that deforestation may be driving malaria infection in monkeys, which could be part of the reason for increased human infection rates.

Johnsons et al.'s study has provided an important step towards better understanding the link between deforestation and the levels of monkey malaria in humans living nearby. Their study provides important insights into how we might find ways of managing the landscape better to reduce health risks from wildlife infection.

## Introduction

Zoonotic infectious diseases arise from the spillover of pathogens into human populations, typically from a reservoir in wildlife hosts. Anthropogenic land use and land cover change have now been widely linked to infectious disease outbreaks (*Brock et al., 2019*; *Davidson et al., 2019a*; *Loh et al., 2016*). Such practices, including deforestation, logging, clearing for cash-crop plantations or conversion of intact forest into arable land, are accelerating across tropical forests of Southeast Asia (*Fornace et al., 2021*; *Imai et al., 2018*; *Fornace et al., 2021*; *Imai et al., 2018*). Mechanisms that underly the association between habitat disturbance and spillover risk from wildlife hosts are complex and occur over multiple spatial scales (*Brock et al., 2019*). In Brazil, re-emergence of Yellow Fever Virus in both NHPs and humans has been linked to areas with highly fragmented forest (*Ilacqua et al., 2021*). In part, an increase in 'edge' habitat in fragmented or mosaic landscapes can facilitate spatial overlap and altered contact patterns between wildlife, vectors, and humans (*Lehman et al., 2006*). Such ecological interfaces are also thought to contribute to parasite spillover in other vector-borne diseases including Zika (*Li et al., 2021a*), Babesiosis and Lyme disease (*Simon et al., 2014*), *Trypanosoma cruzi* (*Vaz et al., 2007*) and zoonotic malaria (*Brock et al., 2019*; *Grigg et al., 2017*). At the same time, habitat fragmentation can have detrimental impact on wildlife population viability, with reduced host species occupancy and reduced disease burden in highly disturbed habitats (*Hanski and Ovaskainen, 2000*). Disentangling this interplay is essential to inform ecological strategies for surveillance and mitigation of diseases in regions undergoing landscape change (*Fornace et al., 2021*).

Zoonotic *P. knowlesi* is a public health threat of increasing importance across Southeast Asia, following the identification of a prominent infection foci in Borneo in 2004 (*Singh et al., 2004*). *P. knowlesi* is a zoonosis, with a sylvatic cycle circulating in non-human primates (NHPs). Human cases currently occur only from spillover events (*Ruiz Cuenca et al., 2022*; *Fornace et al., 2022*; *Fornace et al., 2023*; *Lee et al., 2011*). Human transmission requires bites from infective mosquitos, primarily anopheline mosquitos of the Leucosphyrus Complex (*Anopheles balabacensis, An. latens,*

*An. introlactus*) and Dirus Complex (*An. dirus, An. cracens*) (*Moyes et al., 2016*; *Vythilingam et al., 2006*; *Wong et al., 2015*). Natural hosts for *P. knowlesi* are typically Long-tailed macaques (*Macaca fascicularis*) and Southern Pig-tailed macaques (*M. nemestrina*) (*Moyes et al., 2016*), both occurring widely across Southeast Asia. Currently, distribution of *P. knowlesi* cases is thought to be restricted to the predicted ranges of known vector and host species (*Davidson et al., 2019b*), although recent studies have also identified other NHPs found to be harbouring *P. knowlesi*. This includes Stump-tailed macaques (*M. arctoides),* which are now considered to be another natural reservoir (*Fungfuang et al., 2020*).

Progress towards malaria elimination in Malaysia has been stymied by a recent rise in human incidence of *P. knowlesi* malaria. Even after accounting for increases in surveillance and diagnostic improvements it is now recognised as the most common cause of clinical malaria in Malaysia (*Cooper et al., 2020*). Indeed, Malaysia was the first country not to qualify for malaria elimination due to ongoing presence of zoonotic malaria and the WHO updated the guidelines to reflect zoonotic malaria as a public health threat (*World Health Organization, 2021*). Emergence of *Plasmodium knowlesi* infections has been linked to changes in land cover and land use (*Fornace et al., 2021*). While sporadic cases have been reported across Southeast Asia, including in Indonesia (*Setiadi et al., 2016*), the Philippines (*Fornace et al., 2018*), Vietnam (*Maeno et al., 2015*), Brunei (*Koh et al., 2019*), and Myanmar (*Ghinai et al., 2017*), the majority of *P. knowlesi* cases are found in East Malaysia (Borneo) with hotspots in the states of Sabah and Sarawak (*Jeyaprakasam et al., 2020*), areas that have seen extensive deforestation and landscape modification. In Sabah, human prevalence of *P. knowlesi* infection has recently been shown to be specifically associated with recent loss of intact forest, agricultural activities, and fragmentation across multiple localised spatial scales (*Brock et al., 2019*; *Fornace et al., 2019b*; *Fornace et al., 2016*).

Prevalence of the pathogen in reservoir hosts is one of three crucial factors determining the force of infection in zoonotic spillover events (*Murray and Daszak, 2013*). Despite this, very little is known of the impact of rapid landscape change on the distribution of *P. knowlesi* in NHPs. Literature on the impacts of fragmentation on primates tends to focus on primate density and abundance (*Link et al., 2010*; *Zunino et al., 2007*). What is known is that effects of land cover changes on primate-pathogen dynamics are highly variable and context-specific. Although the vector species responsible for sylvatic transmission remain unknown, the *Anopheles leucospryphus* group, the only vector group implicated in *P. knowlesi* transmission, is widely associated with secondary, disturbed forest (*Brant, 2011*; *Hawkes et al., 2019*; *Wong et al., 2015*). Macaques have been known to preferentially rely on fringe habitat, a behaviour that may be exaggerated in response to habitat fragmentation and facilitate exposure to vectors (*Lehman et al., 2006*; *Stark et al., 2019*). Changes to land composition can also create the biosocial conditions for higher rates of parasitism in primates. Under conditions of limited resources and reduction in viable habitat, conspecific primate density may increase as troops compete for available space. In turn, this can favour transmission via intra-species contact or allow the exchange of pathogens between troops dwelling in interior forest versus edge habitat (*Faust et al., 2018*; *Stark et al., 2019*). Habitat use may also become more intensive, preventing parasite avoidance behaviours (*Nunn and Dokey, 2006*). Land cover change is also known to favour more adaptable, synanthropic species such as *M. fascicularis* (*McFarlane et al., 2012*). Considering the spillover risk posed by wildlife reservoirs of *P. knowlesi,* clarifying any relationships between environmental factors and parasitaemia in key host species may contribute to a more comprehensive understanding of *P. knowlesi* transmission patterns.

Earth Observation (EO) data provides novel opportunities to investigate epidemiological patterns of diseases which are linked to environmental drivers (*Kalluri et al., 2007*). In relation to *P. knowlesi,* utility of fine-scale remote-sensing data has been demonstrated: examples include satellite-derived data used to examine household-level exposure risk in relation to proximate land configuration (*Fornace et al., 2019b*), UAV-imagery used to link real-time deforestation to macaque host behavioural change (*Stark et al., 2019*), and remote-sensing data used to interrogate risk factors for vector breeding sites (*Byrne et al., 2021*). Although macroecological studies that utilise geospatial data are often confounded by issues of matching temporal and spatial scales, as well as by the quality and accuracy of available georeferencing, measures can be taken to account for this when examining the role of environmental factors in modulating disease outcomes. Furthermore, ecological processes occur and interact over a range of distances, or 'spatial scales' (*Brock et al., 2019*; *Fornace et al.,*

*2016*; *Loh et al., 2016*). This applies to determinants of vector-borne disease ecology, from larval breeding microclimate to wildlife host foraging behaviour. As multiple influential variables are rarely captured by a single scale (*Cohen et al., 2016*), data-driven methods can be applied to examine risk factors over multiple scales and identify covariates at their most influential extent (*Byrne et al., 2021*).

We hypothesise that prevalence of *P. knowlesi* in primate host species is spatially heterogeneous and that higher prevalence is partially driven by forest loss and fragmentation, contributing to the strong associations described between land use, land cover and human *P. knowlesi* risk. This study is the first to systematically assess *P. knowlesi* prevalence in NHPs at a regional scale, and across a wide range of habitats. In conceptual frameworks and transmission models, it is often assumed that *P. knowlesi* infections in NHPs are chronic (low level, persistent infection) and ubiquitous (uniformly distributed across populations; *Brock et al., 2016*; *Jeyaprakasam et al., 2020*). No studies have systematically assessed the extent and quality of all available data on *P. knowlesi* in NHPs. Independent studies investigating *P. knowlesi* in primates are typically constrained by small sample sizes and confined geographic areas, limiting inference that can be made about relationships between infection dynamics and landscape characteristics. Systematic tools developed for epidemiological studies of disease prevalence in human populations are rarely applied to the study of wildlife disease prevalence; however, such tools can be used to capture the scale and contrast required in macroecological studies to quantify disease burdens regionally. Furthermore, while recent research has shown the impact of deforestation on the distribution of macaques in the context of *P. knowlesi* (*Moyes et al., 2016*; *Stark et al., 2019*), associations between landscape and variation in the prevalence of simian *Plasmodium* spp. in primates have not been explored. We aimed to (1) assemble a georeferenced dataset of *P. knowlesi* in NHPs; (2) evaluate variation in NHP *P. knowlesi* prevalence by geographic region; and (3) assess environmental and spatial risk factors for *P. knowlesi* prevalence in NHPs across Southeast Asia.

## Results

A systematic literature review was conducted in Medline, Embase, and Web of Science to identify articles reporting prevalence of naturally acquired *Plasmodium knowlesi* in NHPs. Twenty-three research articles were identified (*Akter et al., 2015*; *Amir et al., 2020*; *Chang et al., 2011*; *Fungfuang et al., 2020*; *Gamalo et al., 2019*; *Ho et al., 2010*; *Zamzuri et al., 2020*; *Jeslyn et al., 2011*; *Kaewchot et al., 2022*; *Lee et al., 2011*; *Li et al., 2021b*; *Muehlenbein et al., 2015*; *Nada-Raja et al., 2022*; *Putaporntip et al., 2010*; *Saleh Huddin et al., 2019*; *Salwati et al., 2017*; *Seethamchai et al., 2008*; *Shahar, 2019*; *Gilhooly et al., 2016*; personal correspondence, 2013 and 2015 ; *Vythilingam et al., 2008*; *Yusuf et al., 2022*; *Zhang et al., 2016*), containing 148 unique primate survey records to form the dataset for analyses (see Appendix 2 for details of JBI Critical Assessment) (*Munn et al., 2015*). Year of sampling ranges from 2004 to 2019. No primatological studies were identified from Vietnam, Brunei, or Timor-Leste. Full characteristics of the articles and individual study methodologies are reported in *Appendix 1—table 2*. Spatial resolution of the survey sites varied from GPS point coordinates to country-level administrative boundaries (*Appendix 5—table 1*). Geographic distribution of sampling is illustrated in *Figure 1*.

Overall, records report on a total of 6322 primates, with the largest proportion sampled from Peninsular Malaysia (48.5%, n=3069/6322). Primate surveys were primarily conducted on Long-tailed macaques (*Macaca fascicularis*) (90.5%, n=5720/6322) followed by Pig-tailed macaques (*M. nemestrina*; n=532/6322; *Amir et al., 2020*; *Lee et al., 2011*; *Muehlenbein et al., 2015*; *Putaporntip et al., 2010*; *Appendix 1—table 3*). Reported prevalence of *Plasmodium knowlesi* in NHPs ranged from 0% to 100%. Only 87 of the surveys (58.8%, n=87/148) reported a positive diagnosis, with the remaining 61 sites finding no molecular evidence of *P. knowlesi* infection (41.2%) in any primates tested. A full breakdown of *P. knowlesi* infection rates according to reported primate characteristics can be found in Appendix 1.

### Meta-analysis of *P. knowlesi* prevalence

To quantify regional heterogeneity in simian cases of *P. knowlesi,* a one-stage meta-analysis of prevalence (number positive out of the number sampled) was conducted on primate malaria survey data. Overall pooled estimate for *P. knowlesi* prevalence was 11.99% (CI95% 9.35–15.26). Overall heterogeneity was assessed using the $I^2$ statistic. Substantial between-study heterogeneity ($I^2 \geq 75\%$) was found

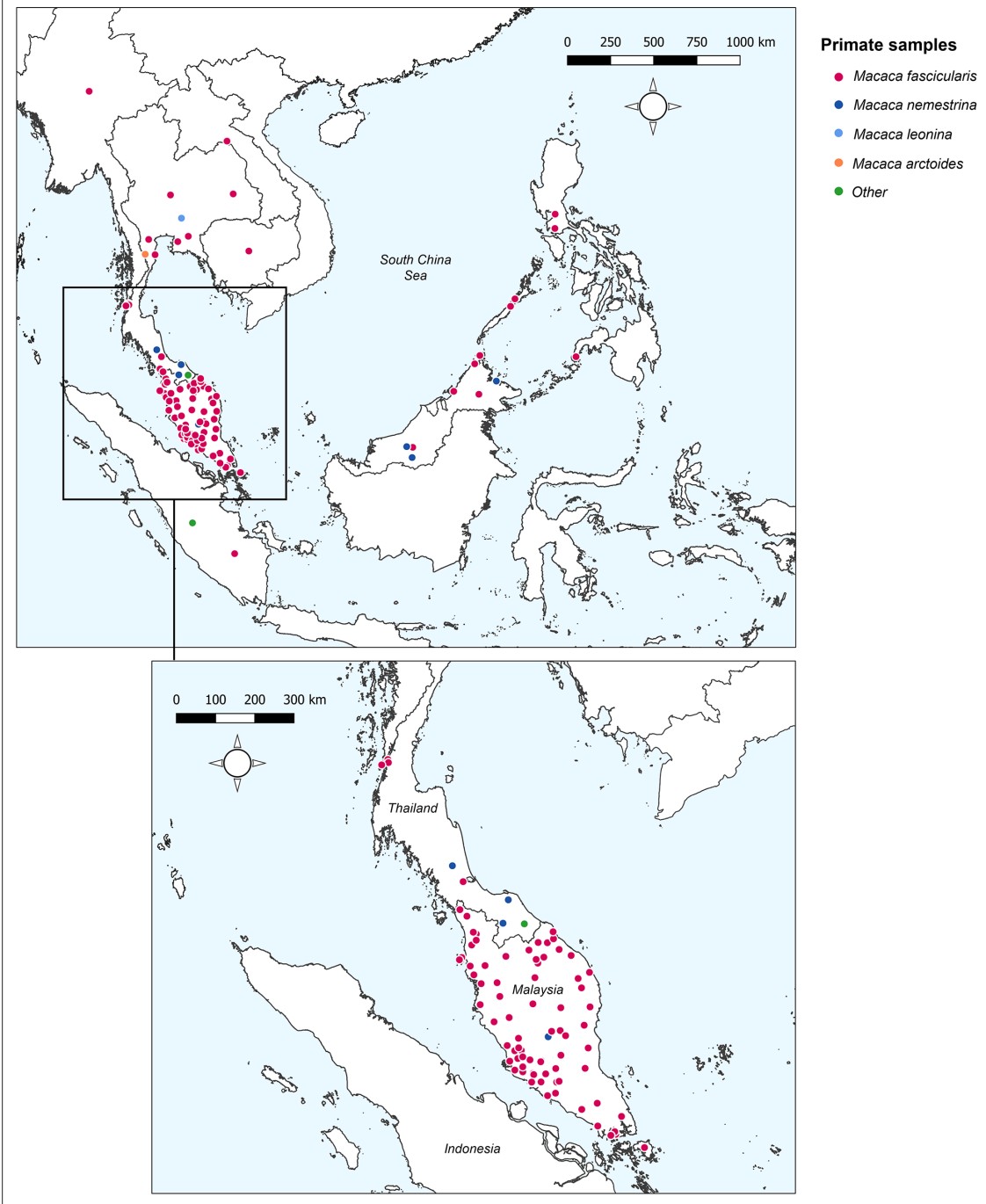

**Figure 1.** Sampling sites and primate species sampled across Southeast Asia. 'Other' includes Trachypithecus obscurus and undefined species from the genus Presbytis. Total surveys (n) = 148.

across all prevalence records ($I^2$=80.5%; CI95% 77.3–83.1). In the sub-group analysis by region, pooled prevalence estimates are consistently low for Thailand (2.0%, CI95% 1.1–3.5%), moderate in Peninsular Malaysia (14.3%, CI95% 11.1–18.2) and elevated in Singapore (23.3%, CI95% 11.0–42.8) and Malaysian Borneo (41.1%, CI95% 20.8–64.9) (*Figure 2*). Sub-group heterogeneity was assessed using prediction intervals, derived from $\tau^2$ statistic used to describe between-study variability. Prediction intervals indicate high heterogeneity of estimates within regions, consistent with expectations of high variability of prevalence across individual study sites. Detailed forest plots for individual prevalence estimates can be found in *Appendix 3—figure 2*.

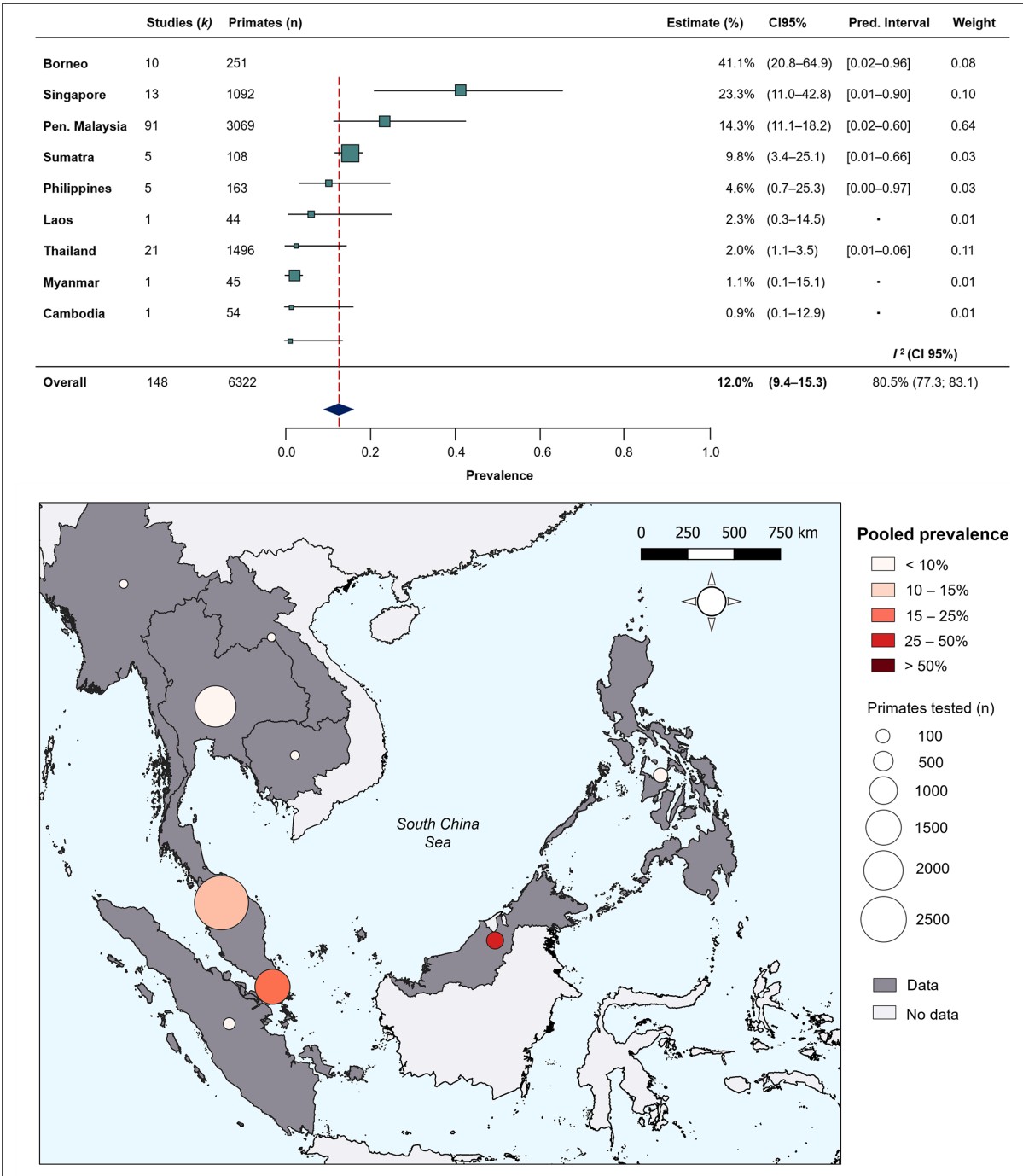

**Figure 2.** Random-effects meta-analysis of *P. knowlesi* prevalence across Southeast Asia. (**A**) Forest plot of pooled estimates for *P. knowlesi* prevalence (%) in all non-human primates tested (n=6322) across Southeast Asia, disaggregated by species and sampling site (k=148). Random-effects meta-analysis sub-grouped by region, with 95% confidence intervals and prediction intervals. (**B**) Map of regional prevalence estimates for *P. knowlesi* prevalence in NHP in Southeast Asia from meta-analysis. Point colour denotes pooled estimate (%). Size denotes total primates tested per region (**n**). Shading indicates data availability.

## Risk factor analysis

Covariate data and *P. knowlesi* prevalence data were used to fit additional models to explore the relationships between localised landscape configuration and NHP malaria prevalence. Environmental covariates were extracted from satellite-derived remote sensing datasets (*Table 1*) at either true sampling sites (GPS coordinates) or 10 random pseudo-sampling sites to account for geographic

**Table 1.** Spatial and temporal resolution (res.) and sources for environmental covariates. Summary metrics extracted within 5, 10 and 20 km circular buffers.

| Covariate | Spatial res. | Temporal res. | Source |
| --- | --- | --- | --- |
| Human density (p/km²) | 1 km | 2012 | *WorldPop, 2018* |
| Elevation (m) | 1 km | 2003 | SRTM 90 m Digital Elevation v4.1 *Jarvis et al., 2008* |
| Tree cover (1/0)* | 30 m | Annual | Hansen's Global Forest Watch *Hansen et al., 2013* |

*Derivatives: Proportion canopy cover (%), Perimeter: area ratio (PARA >0)

uncertainty in prevalence data. Host species was grouped as '*Macaca fascicularis*' or 'Other' due to sample counts of <10 for certain primate species. Only 57.4% (n=85/148 records) of data included year of sampling, deemed to be insufficient to assess temporal patterns in prevalence. Tree canopy cover ranged from negligible to near total cover (100%) within buffer radii (*Appendix 4—table 2*). Details of covariate data processing is illustrated in Appendix 4.

Following a two-stage approach for selection of explanatory variables, tree cover and fragmentation (measured by perimeter: area ratio, PARA) were retained at 5 km as linear terms, human population density was retained at both 5 km and 20 km and primate species was retained as a categorical variable. Spearman's rank tests for residual correlation between final variables at selected scales indicates a strong negative correlation between tree cover and fragmentation index (PARA; $\rho$ = –0.75; *Appendix 6—figure 2*).

Adjusting for all other covariates in the model, we identified strong evidence of an effect between increasing tree canopy cover and higher prevalence of *P. knowlesi* in NHPs within a 5 km radius (aOR = 1.38, CI95% 1.19–1.60; p<0.0001). Evidence was also found for an association between likelihood of *P. knowlesi* and higher degrees of habitat fragmentation (PARA) within 5 km (aOR = 1.17, CI95% 1.02–1.34, p<0.0281). Evidence suggests that human population density within a 5 km radius is associated

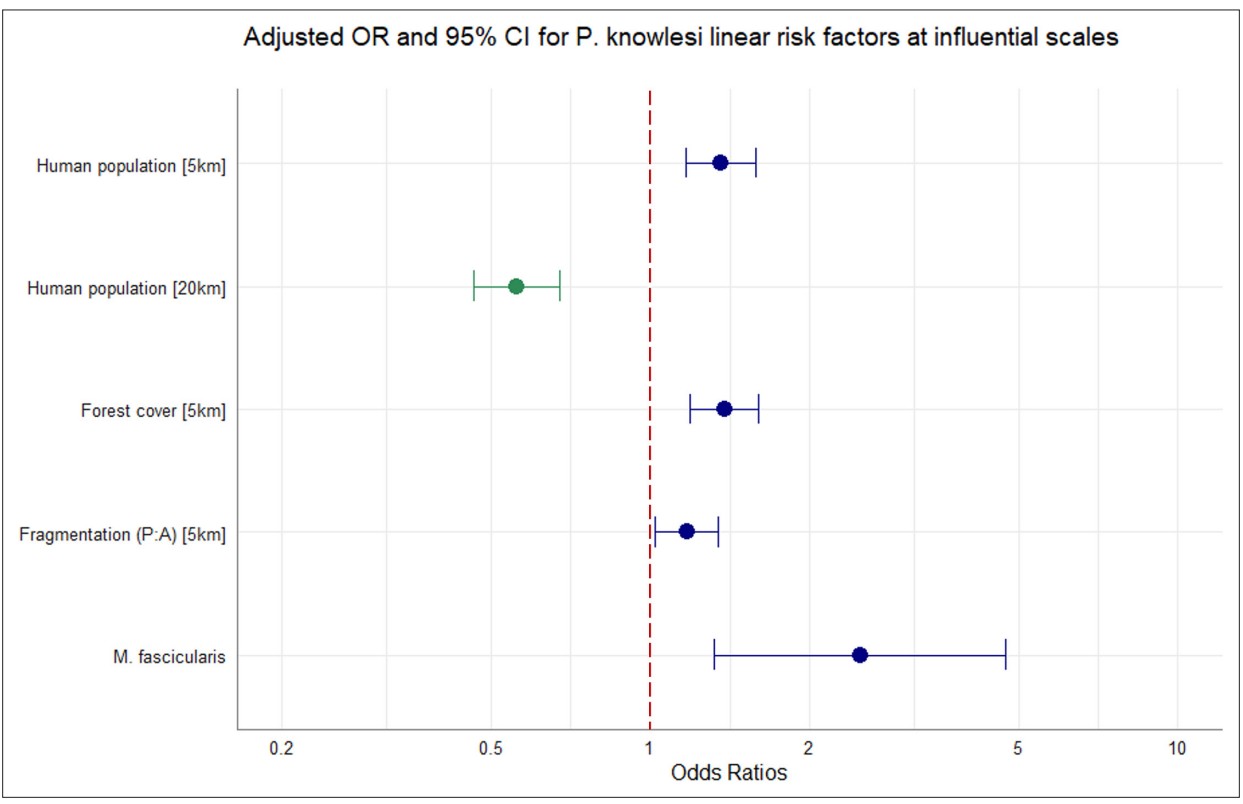

**Figure 3.** Multivariable regression results. Spatial scale denoted in square bracket. Canopy cover = %. Adjusted odds ratios (OR, dots) and 95% confidence intervals (CI95%, whiskers) for factors associated with *P. knowlesi* in NHPs at significant spatial scales. N=1354, accounting for replicate pseudo-sampling.

with risk of *P. knowlesi* in NHP (aOR = 1.36, CI95% 1.16–1.58, p=0.0001) whilst human density within 20 km has an inverse effect on likelihood of *P. knowlesi* (aOR = 0.56, CI95% 0.46–0.67, p<0.0001). *M. fascicularis* is also associated with higher prevalence relative to all other non-human primate species (aOR = 2.50, CI95% 1.31–4.85; p=0.0051). Additional complexity did not improve optimal model fit and effect modification was not pursued. In sensitivity analyses removing data points with excessive spatial uncertainty or restricting data points only to areas with high probability of macaque occurrence, evidence was consistently found that tree canopy cover (5 km) and host species exhibit a strong positive association with prevalence of *P. knowlesi* in NHP (Appendix 6). Final adjusted OR for the full multivariable model can be visualised in *Figure 3*.

## Discussion

Land use and land cover change is widely linked to spillover of zoonotic pathogens from sylvatic reservoirs into human populations, and pathogen prevalence in wildlife host species is key in driving the force of infection in spillover events. Our initial analyses found that for *Plasmodium knowlesi,* there is substantial spatial heterogeneity and prevalence in non-human primates varies markedly between regions of Southeast Asia (*Zhang et al., 2016*). Consistent with our hypothesis that parasite density in primate hosts would be higher in areas experiencing habitat disturbance, we identified strong links between *P. knowlesi* in NHPs and measures of contemporaneous tree cover and habitat fragmentation. To our knowledge, this is the first systematic study to find evidence of landscape influencing the distribution of *P. knowlesi* prevalence in NHPs. Results offer evidence that *P. knowlesi* infection rates in NHPs are linked to changes in landscape across broad spatial scales, and that prevalence of *P. knowlesi* in reservoir species may be driving spillover risk across Southeast Asia. These findings could provide insight to improving surveillance of *P. knowlesi* and to the development of ecologically targeted interventions.

While previous studies have estimated that *P. knowlesi* infection would be chronic in all macaques, or as high as 50–90% for modelling *P. knowlesi* transmission in Malaysia (*Brock et al., 2016*), this data strongly suggests that this is not the case. Overall prevalence of *P. knowlesi* infection in all NHPs is markedly lower than usual estimates, emphasising the importance of accounting for absence data in estimations of prevalence. Considerable heterogeneity was identified between and within regional estimates for *P. knowlesi* across Southeast Asia, which likely reflects genuine differences according to distinct climates and habitats (*Shearer et al., 2016*). Malaysian Borneo was found to have an estimated prevalence over five-fold higher than West Malaysia. Crucially, such extreme prevalence estimates for NHPs in Borneo align with the known hotspot for human incidence of *P. knowlesi* (*Cooper et al., 2020*). By comparison, for Peninsular Malaysia, estimated prevalence is far lower than anticipated. Cases of human *P. knowlesi* do occur in West Malaysia, although transmission has been found to exhibit spatial clustering (*Phang et al., 2020*) which may correspond to pockets of high risk within the wider context of low prevalence of *P. knowlesi* in macaque populations. Regional trends in *P. knowlesi* also mask differences in infection rates between sample locations, driven by more localised factors. Multiple studies reported finding *P. knowlesi* infections in wild macaques to be low or absent in peri-domestic or urbanised areas, attributed to the absence of vector species typically found in forest fringes (*Brant et al., 2016*; *Chua et al., 2019*; *Manin et al., 2016*). This pattern is seen in reports from Peninsular Malaysia (*Saleh Huddin et al., 2019*; *Vythilingam et al., 2008*), Singapore (*Jeslyn et al., 2011*; *Li et al., 2021b*), and Thailand (*Fungfuang et al., 2020*; *Putaporntip et al., 2010*). The high heterogeneity of reports here suggests that the picture is even more complex. *P. knowlesi* infections may even vary between troops within a single study site, as was seen in the Philippines (*Gamalo et al., 2019*). Fine-scale interactions are unlikely to be captured by the scale of this study.

Ecological processes determining *P. knowlesi* infection are influenced by dynamic variables over multiple spatial scales (*Cohen et al., 2016*). We utilised a data-driven methodology to select variables at distances that capture maximum impact on *P. knowlesi* prevalence (*Byrne et al., 2021*; *Fornace et al., 2019b*), with tree cover and fragmentation influential at localised scales and human population density also exerting influence within wider radii. Contrary to previous studies on risk factors for human incidence of *P. knowlesi* (*Fornace et al., 2019b*; *Fornace et al., 2016*), elevation was not found to be associated with *P. knowlesi* in NHPs at any scale. Vector and host species composition vary substantially across tropical ecotones, and it is likely that the study extent encompasses a range of putative vectors across different landscapes, such as those of the Minimus Complex in

northern regions (*Parker et al., 2015*) or the recently incriminated *An.-collessi* and *An.-roperi* from the Umbrosus Group (*De Ang et al., 2021*). Given that the vector species driving sylvatic transmission remain elusive, it is conceivable that the elevation range covers multiple vector and host species niches and explains the lack of observed relationship between elevation and *P. knowlesi* in NHPs. Human population density was found to be significant at multiple distances, with contrasting effects on parasite prevalence in NHP. Previous studies have found a negative association between human density and vector density and biting rates in forested landscapes (*Fornace et al., 2019a*). Across wide spatial scales, increased vector density in less populated, more forested areas could generate higher parasite prevalence in NHPs. At the same time Long-tailed macaques, a species shown here to have higher prevalence rates, are notorious as nuisance animals and many of the available samples were collected opportunistically in urban areas, which might underly the observed positive association between localised high human density and higher prevalence in NHP. Whilst more data would be needed to understand this interaction, this further demonstrates the importance of using approaches to identify disease dynamics across multiple spatial scales (*Brock et al., 2019*).

A key finding is the link between high prevalence of *P. knowlesi* in primate host species with high degrees of habitat fragmentation. Habitat fragmentation is a key aspect of landscape modification, where large contiguous areas of habitat (for example, forests) are broken into a mosaic of smaller patches. This disturbs the ecological structure by increasing the density of fringes or 'edges', dynamic habitat often at the boundaries between natural ecosystems and human-modified landscapes (*Borremans et al., 2019*). Other studies have linked habitat fragmentation to increased generalist parasite density in primates. In Uganda, a higher prevalence and infection risk of protozoal parasites was observed in wild populations of red colobus primates (*Procolobus rufomitratus*) inhabiting fragmented forests compared to those in undisturbed habitat (*Gillespie and Chapman, 2008*). For *P. knowlesi,* creation of edge habitat is thought to favour vectors of the Leucosphyrus Complex (*Davidson et al., 2019a*; *Hawkes et al., 2019*). *Anopheles* spp. presence can be predicted by indices of fragmentation in Sabah, Borneo, with land cover changes creating more suitable micro-climate for larval habitats (*Byrne et al., 2021*), and an increased abundance of *An. balabacensis* found in forest fringes (*Hawkes et al., 2019*; *Wong et al., 2015*). Increasing landscape complexity results in increased density of edge habitat, with conceivably higher density of vectors in forest fringes. Therefore, preferential use of fringe habitat and high exposure to vectors in forest fringes may contribute to higher conspecific transmission of *P. knowlesi* between primates in increasingly fragmented habitats. This finding also lends clarity to landscape fragmentation as a risk factor for human exposure to *P. knowlesi* in Malaysian Borneo (*Brock et al., 2019*; *Fornace et al., 2019b*), with changes in relative host density, vector density and wildlife parasite prevalence in nascent forest fringes potentially enhancing the spillover of this disease system into human populations in fragmented habitats.

Conversely, we saw a strong association between high parasite prevalence and high tree canopy coverage. Given that a strong inverse relationship with fragmentation was observed, with high tree density correlating to low fragmentation indices and vice versa, this speaks to a trade-off between dense canopy cover and high habitat complexity and suggests an 'ideal' amount of habitat fragmentation that facilitates prevalence in primate hosts. For animals with larger home ranges, individual-based disease models combined with movement ecology approaches have shown that the most highly fragmented areas are less favourable for maintaining parasite transmission (*White et al., 2018*). In Sabah, individual macaques were shown to increase ranging behaviour in response to deforestation (*Stark et al., 2019*). Forest edge density also peaks at intermediate levels of land conversion (*Borremans et al., 2019*). With smaller habitat patches in maximally fragmented landscapes potentially insufficient to support macaque troops, this interplay between disease ecology and metapopulation theory may explain why both tree density and habitat fragmentation appear to pose a greater risk for simian *P. knowlesi*. Likewise, this may relate to the finding that in Borneo, larger forest patches (lower fragmentation indices) were associated with *P. knowlesi* spillover in Borneo (*Fornace et al., 2019b*). Overall, this finding offers an insight to mechanisms that underpin the increased force of infection of *P. knowlesi* that is associated with landscape change.

There are limitations to consider in the available data and interpretation of these findings. 'Small-study effects' were observed in the dataset, suggestive of a bias toward positive effect estimates (*Stewart et al., 2012*). This may be a result of data disaggregation and small studies creating artefactually higher estimates or may reflect true bias in data collection toward areas known to be endemic

for *P. knowlesi* and convenience sampling of macaques. Assumptions have also been made that sample site equates to habitat, which may not reflect actual habitat use, and even accurate georeferenced data points are unlikely to entirely reflect surrounding habitat within the macaque home range. Variability in study designs and data reporting also impacted geospatial accuracy. Steps were taken to account for spatial bias by extracting covariates at randomly generated pseudo-sampling points. Whilst uncertainty cannot be eliminated, we demonstrate a robust methodology to accommodate for geographical uncertainty in ecological studies. Future investigations should prioritise systematic, georeferenced sampling across a range of landscape scenarios.

Results show important regional ecological trends, but broad geographic patterns may not be generalisable at individual levels, or to all putative host species in all geographic contexts (*Zhang et al., 2016*). Follow-up studies should be conducted at higher spatial and temporal resolution to characterise the effect of local landscape configuration on wildlife *P. knowlesi* prevalence. Effects of fragmentation are likely to be dependent on land conversion type, species composition and surrounding matrix habitat (*Fornace et al., 2019b*). Use of perimeter: area ratio (PARA) as a fragmentation index was justified given high canopy coverage in study sites (*Wang et al., 2014*), although Edge Density (ED) or normalised Landscape Shape Index (nLSI) might be more appropriate in future analyses to account for variation in forest abundance. Specific land configurations have previously been linked to *P. knowlesi* exposure in Borneo (*Fornace et al., 2019b*), notably in areas where palm oil plantation is a dominant industry. Given this, broad forest classifications used here may mask important differences in *P. knowlesi* prevalence between land classes. As it was not possible to include contemporary land cover classifications in this analysis, future studies would also benefit from looking at specific habitat type (e.g., primary forest, agroforest, plantation).

## Concluding remarks

Strong links have been identified between land use and land cover change and ecosystem perturbation that favours the transmission of vector-borne diseases (*Loh et al., 2016*). Prevalence of *P. knowlesi* in macaques is likely to be a crucial determinant of human infection risk, and more representative estimates of *P. knowlesi* prevalence derived here can better inform regional transmission risk models. This study also characterises landscape risk factors for heightened prevalence of *P. knowlesi* in NHPs. Findings provide evidence that *P. knowlesi* in primate hosts is partly driven by landscape modification across Southeast Asia. While the full complexity is not captured by the covariates used, it is clear that *P. knowlesi* infection in NHPs is not restricted to densely forested areas. This study also demonstrates the utility of systematic meta-analysis tools and remote-sensing datasets in the investigation of macroecological disease trends, in conjunction with methods to standardise a spatially heterogeneous dataset and data-driven selection of spatial scales. Gaps identified in data reporting should inform more systematic and localised primatological surveys to disentangle precise mechanisms. Notwithstanding limitations, this study highlights the marked spatial heterogeneity and role of landscape complexity in driving *P. knowlesi* infection rates in NHPs. Given the clear intersection between human epidemiology and wildlife ecology, it is essential that infection dynamics within wildlife reservoirs are considered in future public health interventions.

## Methods

### Study site

This study focused on the simian malaria *Plasmodium knowlesi* across Southeast Asia, within 28°30'00.0"N, 92°12'00.0"E and 11°00'00.0"S, 141°00'00.0"E. Climate mainly corresponds to the equatorial tropical zone, with high temperatures and high humidity.

### Data assembly

A systematic literature review was conducted under the CoCoPop framework (Condition, Context, Population) (*Ruiz Cuenca et al., 2022*; *Munn et al., 2015*). All studies identified in the literature review were screened for data on NHPs with a confirmed *P. knowlesi* diagnosis or absence data (zero counts of *P. knowlesi* with appropriate diagnostic methods). Exclusion criteria included (a) studies exclusively relying on microscopy (*Antinori et al., 2013*) (b) laboratory, animal model or experimental infection studies (c) data from outside of Southeast Asia. No limit was set on the temporal range for

primate survey records. Duplicate records reporting results from the same surveys were removed, with one record per survey retained. Critical appraisal of the studies was conducted using the Joanna Briggs Institute (JBI) checklist for prevalence studies (*Munn et al., 2015*; see Appendix 1 for details and criteria). A flowchart of the selection process is illustrated in *Appendix 1—figure 3*, with a full list of articles included provided in *Appendix 1—table 2*.

Primary outcome was defined as *P. knowlesi* prevalence (*p*, proportion positive for *P. knowlesi* infection from *n* sampled NHPs). For each independent primate study, the following variables were extracted: year of data collection, primate species sampled, primate status (wild/captive), diagnostic test (PCR/sequencing) and target gene(s), sampling method (routine/purposive), number of *P. knowlesi* positive samples, number of *Plasmodium* spp. positive samples, total number of primates tested and geographical information.

In most studies identified, study site was only geolocated to a geographic area or descriptive location. Geolocation was assigned at the lowest available level of administrative polygon (i.e. district/state/country) by cross-referencing reported sampling location with GADM (v3.6) administrative boundaries. If specific location was given, GPS coordinates were assigned via Google Maps. For data visualisation, point coordinates were plotted in QGIS (3.10.14) and R (4.1.0) software.

## Meta-analysis of *P. knowlesi* prevalence

Meta-analysis was conducted using methods that are standard in the analysis of human disease prevalence for individual participant datasets (IDP) (*Liberati et al., 2009*; *Stewart et al., 2012*). Data were disaggregated by geographic location (site) and primate species, to illustrate variance in prevalence by survey unit (*Stewart et al., 2012*). One-stage meta-analysis is considered appropriate for studies where the outcome may be infrequent, so data was included in a single model under the 'DerSimonian and Laird' variance estimator (*Munn et al., 2015*). Sensitivity analyses were conducted to compare methods for the back-transformation of prevalence estimates. For studies where prevalence estimates tend towards 0% or 100%, variance tends towards 0. To stabilise the variance and enable back-transformation of zero prevalence records, logit method was selected for the transformation of prevalence, with the inverse variance method used for individual study weights (see Appendix 3 for details).

Overall heterogeneity of prevalence records was assessed using the $I^2$ statistic (*von Hippel, 2015*), a relative estimate of true between-study variance. Sub-group analysis was conducted according to geographic region, with the heterogeneity of reported prevalence within regional sub-groups assessed using prediction intervals derived from the $\tau^2$ statistic. Small-study effects, including selection and publication biases, were assessed by examining funnel plots and imputing 'missing' estimates using the trim-and-fill method (*Lin and Chu, 2018*). Full rationale and details of small-study effect assessments can be found in Appendix 3.

## Remote sensing data

Satellite-derived remote sensing datasets were used to assemble local environmental and anthropogenic covariates. Gridded UN-adjusted human population estimates were assembled at 1 km resolution from *WorldPop, 2018*. Elevation data was obtained from NASA SRTM 90 m Digital Elevation Database v4.1 (CGIAR-CSI) (*Jarvis et al., 2008*) with a spatial resolution of 1 km. Contemporaneous tree cover was derived from Hanson's Global Forest Watch (30 m) (*Hansen et al., 2013*), extracted for every year between 2006 and 2020.Tree cover was classified as ≥50% crown density, and then matched to primate data by sample site geolocation and by year of sample collection to account for rapid forest loss (*Appendix 4—figure 1*). Where a broad timeframe of sampling was provided (≥3 years), median year was used. Full details for variable selection and processing can be found in Appendix 4.

Perimeter: area ratio (PARA, ratio of patch perimeter length to patch surface area) of given land class is a key metric for habitat conversion, where a higher PARA provides a measure of boundary complexity and indicates a more fragmented landscape (*McGarigal et al., 2021*). Mean PARA was extracted from canopy cover within circular buffers. Habitat fragmentation has been shown to correlate with disease transmission parameters (*Borremans et al., 2019*; *Faust et al., 2018*), but definitions often lack precision and can be considered with respect to 'separation effects' (division and isolation of patches) and 'geometric effects' (changes to ratios of perimeter and core habitat;

*Wilkinson et al., 2018*). PARA provides a measure of edge density within the buffer area (PARA >0) and has been shown to provide a good index of fragmentation and good discrimination of spatial aggregation across areas where habitat abundance (tree canopy cover) is high (*Wang et al., 2014*; *Appendix 4—table 2*, *Appendix 4—figure 4*).

## Covariate assembly

For studies with exact GPS coordinates, precise environmental data at a single site could be obtained. For surveys published without GPS coordinates, there is considerable geographic uncertainty in the exact sampling location (Appendix 5). Uncertainty in the spatial and environmental determinants of prevalence generates a sampling bias, with the precision of covariates correlated to certain studies. Use of a single centroid proxy site is standard procedure, but often generates erroneous estimates in large or heterogenous sampling units (*Cheng et al., 2021*). Alternative strategies were employed to account for and mitigate the effect of spatial uncertainty and spatial bias. Each prevalence observation was replicated and assigned a random sample of environmental realisations. 10 random sampling points were generated within the sampling area provided by the study, and covariates were extracted at each proxy sampling site (*Appendix 5—figure 1*). Selection of random points was validated by visual inspection of the stability of model coefficients with the inclusion of an increasing number of points. Number of points was selected conservatively at the point where coefficients stabilised (n=10).

For every georeferenced sampling point, mean values for all selected covariates were extracted within buffer radii at 5 km, 10 km, and 20 km (Appendix 4). Buffer area sizes were selected to investigate multiple spatial scales over which associations between risk factors and *P. knowlesi* prevalence might occur. A minimum radius of 5 km was chosen to approximate the maximum ranging distance for *M. fascicularis* (*Waxman et al., 2014*), with wider radii (10–20 km) included to account for the geographic uncertainties in areal data. Flowchart of data processing chain can be found in *Appendix 4—figure 2*.

## Analysis of environmental risk factors

Generalised linear mixed-effect regression models (GLMM) were fitted to NHP prevalence data using a binomial distribution with a logit link. To account for within-study correlation in reported average prevalence, a unique identifier combining author and study was included as a random intercept in all models. Artificial inflation of sample size in the replicated data (10 pseudo-sampling sites for data geolocated to administrative areas) was accommodated by reducing individual observation weights to 1/10th within the model.

Each covariate at each spatial scale was assessed for inclusion in the multivariable model based on bivariable analysis and a criterion of p>0.2 under likelihood ratio tests (LRT; *Appendix 6—table 1*). A quadratic term for the fragmentation index 'PARA' was included to account for possible nonlinearity. Multicollinearity among independent predictors at multiple scales was examined via variance inflation factors (VIF). The VIF of each predictor variable was examined following a stepwise procedure, starting with a saturated model and sequentially excluding the variable with the highest VIF score from the model. Stepwise selection continued in this manner until the entire subset of explanatory variables in the global model satisfied a moderately conservative threshold of VIF ≤6 (*Rogerson, 2001*). Qualifying variables obtained were then assessed for model inclusion using a backward stepwise strategy, removing variables with the highest *p* value (LRT) until a pre-defined threshold of α<0.05. Spearman's rank tests were conducted on the selected variables to observe residual correlation, plotted as a correlation matrix (*Appendix 6—figure 2*).

Fully adjusted odds ratios (OR) for associations between environmental covariates and *P. knowlesi* prevalence were derived from the final multivariable GLMM with p values derived from LRT. Spatial sensitivity analyses were conducted by excluding data points from administrative boundaries outside a reasonable size or above a reasonable threshold of environmental certainty, according to the standard deviation (SD) of the covariate values within each set of 10 environmental realisations (*Appendix 5— figure 2*, *Appendix 6—tables 4–6*). Ecological sensitivity analyses were conducted by removing data points that fall outside areas with high predicted probability of occurrence for *Macaca fascicularis*,

*Macaca nemestrina,* and *Macaca leonina* and running regression analyses on the constrained dataset (*Moyes et al., 2016*; *Appendix 6—figures 3–5*, *Appendix 6—tables 7–9*).

## Acknowledgements

Research was supported by the Sir Henry Dale Fellowship, jointly funded by the Wellcome Trust and the Royal Society (Grant Number 221963/Z/20/Z). Data collected in Peninsular Malaysia by the Faculty of Veterinary Medicine, Universiti Putra Malaysia, was funded by the Ministry of Higher Education Malaysia (Grant LRGS/1/2018/UM/01/1/). Additional funding was supported by the World Health Organization.

## Additional information

### Funding

| Funder | Grant reference number | Author |
| --- | --- | --- |
| Wellcome Trust | 221963/Z/20/Z | Kimberly Fornace |
| Royal Society | 221963/Z/20/Z | Kimberly Fornace |
| Ministry of Higher Education Malaysia | LRGS/1/2018/UM/01/1 | Reuben Sunil Kumar Sharma |
| World Health Organization | | Kimberly Fornace Chris Drakeley |

The funders had no role in study design, data collection and interpretation, or the decision to submit the work for publication. For the purpose of Open Access, the authors have applied a CC BY public copyright license to any Author Accepted Manuscript version arising from this submission.

### Author contributions

Emilia Johnson, Conceptualization, Data curation, Formal analysis, Validation, Investigation, Visualization, Methodology, Writing - original draft, Writing – review and editing; Reuben Sunil Kumar Sharma, Conceptualization, Resources, Data curation, Supervision, Writing – review and editing; Pablo Ruiz Cuenca, Data curation, Supervision, Validation, Investigation, Methodology, Writing – review and editing; Isabel Byrne, Formal analysis, Methodology, Writing – review and editing; Milena Salgado-Lynn, Conceptualization, Resources, Writing – review and editing; Zarith Suraya Shahar, Lee Col Lin, Norhadila Zulkifli, Nor Dilaila Mohd Saidi, Resources, Data curation; Chris Drakeley, Conceptualization, Funding acquisition, Investigation, Methodology, Writing – review and editing; Jason Matthiopoulos, Conceptualization, Formal analysis, Validation, Investigation, Methodology, Writing – review and editing; Luca Nelli, Formal analysis, Validation, Investigation, Visualization, Methodology; Kimberly Fornace, Conceptualization, Resources, Formal analysis, Supervision, Funding acquisition, Validation, Investigation, Methodology, Project administration, Writing – review and editing

### Author ORCIDs

Emilia Johnson (iD) http://orcid.org/0000-0001-5460-1715
Pablo Ruiz Cuenca (iD) http://orcid.org/0000-0002-2180-9509
Isabel Byrne (iD) http://orcid.org/0000-0002-7800-3733
Milena Salgado-Lynn (iD) http://orcid.org/0000-0003-1769-6465
Chris Drakeley (iD) https://orcid.org/0000-0003-4863-075X
Jason Matthiopoulos (iD) http://orcid.org/0000-0003-3639-8172
Luca Nelli (iD) http://orcid.org/0000-0001-6091-4072
Kimberly Fornace (iD) http://orcid.org/0000-0002-5484-241X

Reviewer #1 (Public Review): https://doi.org/10.7554/eLife.88616.4.sa1
Reviewer #2 (Public Review): https://doi.org/10.7554/eLife.88616.4.sa2
Author response https://doi.org/10.7554/eLife.88616.4.sa3

## Additional files

### Supplementary files

• MDAR checklist

### Data availability

All data of P. knowlesi prevalence in non-human primates were sourced from systematic literature review of publicly available research articles. Published primate datasets are available in *Ruiz Cuenca et al., 2022* and *Shahar, 2019*. Full reference list for all publicly available primate infection data including details of individual articles and data types are available in Appendix 1. Earth Observation (EO) data were derived from open-source environmental datasets. References, links and details of scale and resolution and manipulation are provided in manuscript (Table 1 in Results and details in Methods) with further information provided in the Appendix 4. Example code for manipulation of environmental data and data analysis are available on an associated GitHub repository (copy archived at *Johnson, 2024*).

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

# Appendix 1

## Data assembly

Prior to conducting the study, a review of current literature was constructed to find articles related to '*Plasmodium knowlesi*' or to both 'malaria' and 'primate', including synonyms and sub-headings. The search was elaborated to specify environmental factors (*Appendix 1—figure 1*). The following databases were searched:

- Medline
- Embase
- Web of Science

Provisional data were extracted using a standardised form (*Appendix 1—figure 2*) using standardised definitions (*Appendix 1—table 1*), from which an initial set of studies were identified for this investigation. Duplicate records (confirmed/suspected to be the same specimens) were removed, with one record retained (*Appendix 1—figure 3*)

Embase Classic+Embase <1947 to 2021 July 02>

| | | |
|---|---|---|
| 1 | (monkey* or macaca* or macaque* or primate* or zoono* or simian).mp. | 284859 |
| 2 | Zoonosis/ep, et [Epidemiology, Etiology] | 3000 |
| 3 | Catarrhini/ or Cercopithecidae/ or Cercopithecinae/ or Macaca/ | 29691 |
| 4 | 1 or 2 or 3 | 285484 |
| 5 | Malaria/ep [Epidemiology] | 11117 |
| 6 | (malaria or plasmodium or inui or cynomolgi or coatneyi or knowlesi).mp. | 148966 |
| 7 | 5 or 6 | 148966 |
| 8 | 4 and 7 | 4577 |
| 9 | Plasmodium knowlesi/ | 1349 |
| 10 | plasmodium knowlesi.mp. | 1734 |
| 11 | 9 or 10 | 1734 |
| 12 | 8 or 11 | 5439 |
| 13 | (forest* or biodivers* or ecolog* or fragment* or deforest* or anthropogenic or environment* or climat*).mp. | 2514624 |
| 14 | 12 and 13 | 710 |

**Appendix 1—figure 1.** Search strategy for background research.

| Species | Presence | # | Date | Country | Location | Latitude | Longitude | Source ID |
|---|---|---|---|---|---|---|---|---|
| | | | | | | | | |
| | | | | | | | | |
| | | | | | | | | |
| | | | | | | | | |

**Appendix 1—figure 2.** WHO Report primate data extraction form.

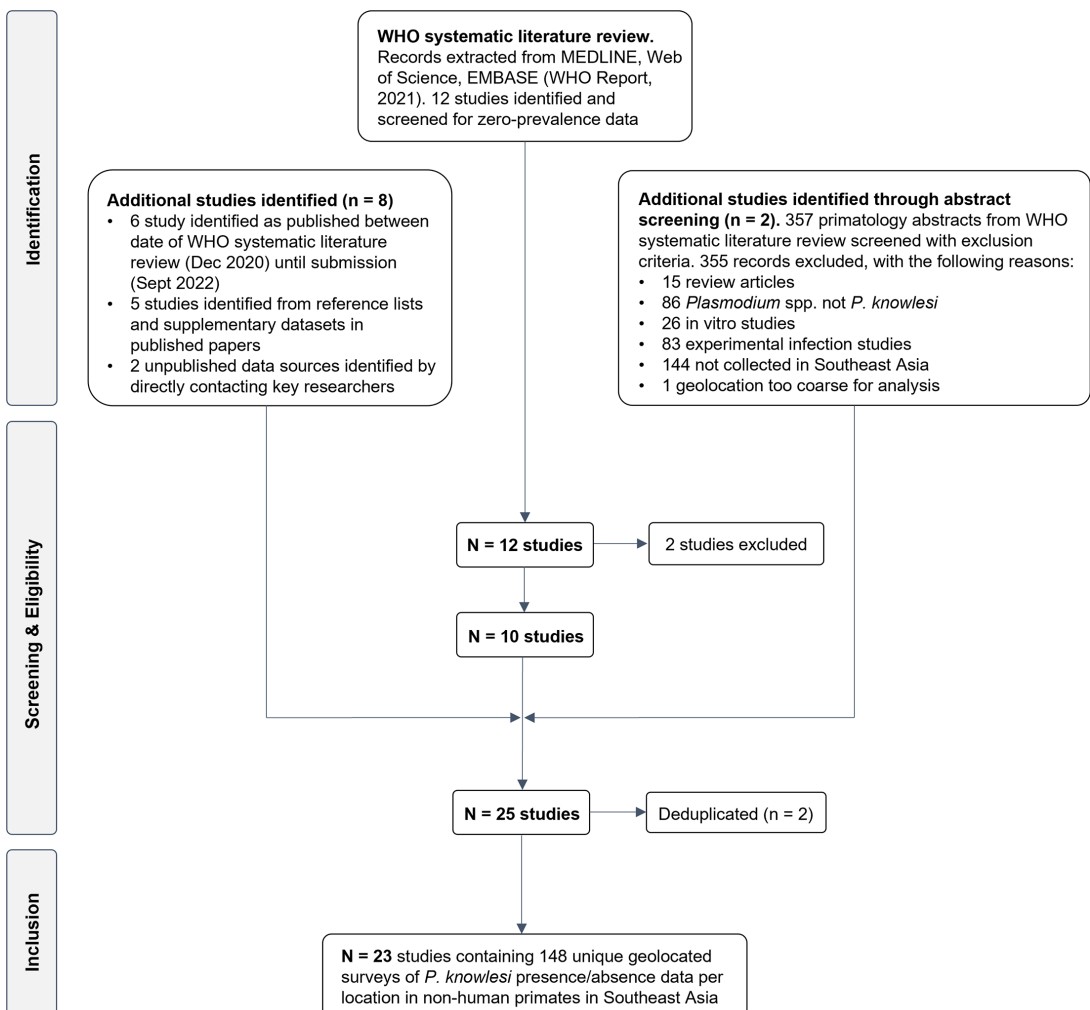

**Appendix 1—figure 3.** Flow chart illustrating study selection process.

**Appendix 1—table 1.** Standardised definitions for qualitative primate characteristics.

| Variable | Category | Definition |
|---|---|---|
| Sampling | Routine | Animals collected for surveillance purposes or extracted from human–conflict zones; data collected opportunistically *Shearer et al., 2016* |
| | Study | Animals captured and sampled specifically for a study of *Plasmodium spp* and/or *P. knowlesi* |
| Status | Captive | Animal resident in sanctuary or conservation park |
| | Wild | Free-living animal, not registered/resident in any sanctuary |
| | Sanctuary | A wildlife sanctuary/rehabilitation centre housing key primate species *Gamalo et al., 2019* |
| Area | Forest | Areas that are uninhabited with extensive tree cover |
| | Peri–domestic | As defined by the author. Example definitions as follows:<br>• Rural areas (areas with low human density, close to secondary/scrub forest) *Shahar, 2019*<br>• Public nature reserve park *Jeslyn et al., 2011*<br>• 2 km from longhouse communities *Lee et al., 2011*<br>• Wild Long-tailed macaque samples collected based on their proximity to humans *Saleh Huddin et al., 2019* |
| | Agricultural | Animal located in agricultural areas, predominantly monoculture (e.g. orchard, plantation) *Shahar, 2019* |
| | Urban | As defined by the author. Generally, areas with high human population density *Li et al., 2021b* |

**Appendix 1—table 2.** Characteristics of the included studies.

| Author | Year(s) | Country/region | N* | Sample† | Diagnostic | Target gene(s) | Primer |
|---|---|---|---|---|---|---|---|
| *Lee et al., 2011* | 2004–2008 | Malaysia/Borneo | 108 | Study | Nested PCR | SSU-rRNA/csp/mtDNA | Kn1f/Kn3r |
| *Seethamchai et al., 2008* | 2006 | Thailand | 99 | Study | Sequencing | A-type-SSU-rRNA/cytb | • |
| *Vythilingam et al., 2008* | 2007 | Malaysia/Peninsular | 145 | Study | PCR/Sequencing | SSU-rRNA/csp | Pmk8/Pmkr9 |
| | 2007 | Singapore | 40 | Study | PCR | • | • |
| | 2007–2010 | Indonesia/Sumatra | 70 | Study | PCR | • | • |
| | 2011 | Cambodia | 54 | Study | PCR | • | • |
| | 2012 | Philippines | 68 | Study | PCR | • | • |
| *Zhang et al., 2016* | 2015 | Laos | 44 | Study | PCR/Sequencing | SSU-rRNA | PK18SF/PK18SRc |
| *Jeslyn et al., 2011* | 2008 | Singapore | 13 | Routine | PCR/Sequencing | SSU-rRNA /csp | Pmk8/Pmkr9 |
| *Ho et al., 2010* | 2008 | Malaysia/Peninsular | 107 | Routine | Nested PCR | SSU-rRNA | Pmk8/Pmkr9 |
| *Li et al., 2021b* | 2008–2017 | Singapore | 1039 | Routine | Nested PCR | SSU-rRNA | Pmk8/Pmkr9 |
| *Putaporntip et al., 2010* | 2009 | Thailand | 655 | Study | Sequencing | cytb | • |
| *Chang et al., 2011* | 2010 | Myanmar | 45 | Study | PCR | SSU-rRNA | • |
| *Muehlenbein et al., 2015* | 2010 | Malaysia/Borneo | 41 | Study | PCR | mtDNA/AMA-1/MSP-1 | • |
| *Shahar, 2019* ‡ | 2010–2017 | Malaysia/Peninsular | 1587 | Routine | Nested PCR | SSU-rRNA | • |
| § | 2013 | Malaysia/Peninsular | 15 | Study | PCR | • | • |
| § | 2013–2016 | Malaysia/Borneo | 25 | Study | Nested PCR | cytb | PKCBF/PKCBR |
| *Saleh Huddin et al., 2019* | 2014 | Malaysia/Peninsular | 415 | Study | PCR/Sequencing | SSU-rRNA | Pmk8/Pmkr9 |
| *Akter et al., 2015* | 2015 | Malaysia/Peninsular | 70 | Routine | PCR/Sequencing | A-type-SSU-rRNA | Pmk8/Pmkr9 |
| *Amir et al., 2020* | 2016 | Malaysia/Peninsular | 103 | Routine | Nested PCR | SSU-rRNA | PkF1140/PkR1550 |
| *Gamalo et al., 2019* | 2017 | Philippines | 95 | Study | Nested PCR | SSU-rRNA | Kn1f/Kn3r |
| *Fungfuang et al., 2020* | 2017–2019 | Thailand | 93 | Study | Nested PCR | SSU-rRNA | Kn1f/Kn3r |
| *Nada-Raja et al., 2022* | 2018 | Malaysia/Borneo | 73 | Study | Nested PCR | SSU-rRNA/csp/mtDNA | Kn1f/Kn3r |

*Appendix 1—table 2 Continued on next page*

*Appendix 1—table 2 Continued*

| Author | Year(s) | Country/region | N* | Sample† | Diagnostic | Target gene(s) | Primer |
|--------|---------|----------------|-----|---------|------------|----------------|--------|
| *Yusuf et al., 2022* | 2016–2019 | Malaysia | 419 | Study | Nested PCR | SSU-rRNA | Kn1f/Kn3r |
| *Zamzuri et al., 2020* | 2018 | Malaysia/Peninsular | 212 | Routine | PCR | • | • |
| *Kaewchot et al., 2022* | 2019 | Thailand | 649 | Study | Nested PCR | SSU-rRNA | Pmk8/Pmkr9 |
| *Salwati et al., 2017* | 2015 | Indonesia/Sumatra | 38 | Study | PCR/Sequencing | • | • |

*N=number of primates sampled.

†Animal trapped either on routine or study basis.

‡Unpublished, personal correspondence (p/c).

§Danau Girang Field Centre, p/c from Dr Salgado Lynn.

Of the 87 records reporting presence of *P. knowlesi,* only 22 records (containing 248 *P. knowlesi* positive macaques) report whether *P. knowlesi* infection was a mono-infection or mixed infection with other simian *Plasmodium* spp. With a low proportion of data represented, this was deemed insufficient to conduct any further investigations.

*Macaca fascicularis* is the predominant species tested. However, reports also include *M. nemestrina* (6.1%, n=9/148; 527 macaques) (*Amir et al., 2020*; *Lee et al., 2011*; *Putaporntip et al., 2010*; *Muehlenbein et al., 2015*), *M. arctoides* (1.4%, n=2/148; 36 macaques) (*Fungfuang et al., 2020*; *Putaporntip et al., 2010*), *M.-leonina* (n=1/148; 25 macaques) (*Fungfuang et al., 2020*), *Trachypithecus obscurus* (Dusky leaf monkey) (n=1/148; 7 tested) and unspecified species from the *Presbytis* genus (n=1/148; 2 tested) (*Appendix 1—table 3*). One study additionally sampled 1 *Presbytis melalophos* (Black-crested Sumatran langur) (*Vythilingam et al., 2008*), but species-specific *P. knowlesi* was not reported (*Appendix 1—table 4*).

**Appendix 1—table 3.** Published studies of *P. knowlesi* infections (n) in non-human primate species collected (N) in Southeast Asia, grouped by region and author.

| Region | Species | | | | Total | Ref |
|--------|---------|---|---|---|-------|-----|
| | *M. fascicularis* | *M. nemestrina* | *M. arctoides* | Other | | |
| Peninsular | 25/107 | • | • | • | | *Ho et al., 2010* |
| Malaysia | 48/415 | • | • | • | | *Saleh Huddin et al., 2019* |
| | 21/70 | • | • | • | | *Akter et al., 2015* |
| | 11/98 | 0/5 | • | • | | *Amir et al., 2020* |
| | 0/15 | • | • | • | | |
| | 10/145 | • | • | • | | *Vythilingam et al., 2008* |
| | 215/1587 | • | • | • | | *Shahar, 2019* |
| | 66/415 | • | • | • | | *Yusuf et al., 2022* |
| | 74/207 | 3/5 | • | • | 473/3069 | *Zamzuri et al., 2020* |
| Borneo | 4/26 | 2/15 | • | • | | *Muehlenbein et al., 2015* |
| | 71/82 | 13/26 | • | • | | *Lee et al., 2011* |
| | 18/25 | • | • | • | | |
| | 7/45 | 2/28 | • | • | | *Nada-Raja et al., 2022* |
| | 2/4 | • | • | • | 119/251 | *Yusuf et al., 2022* |
| Sumatra | 0/70 | • | • | • | | *Zhang et al., 2016* |
| | 5/32 | 1/4 | • | 0/2† | 6/108 | *Salwati et al., 2017* |

*Appendix 1—table 3 Continued on next page*

*Appendix 1—table 3 Continued*

| | Species | | | | | |
| Region | M. fascicularis | M. nemestrina | M. arctoides | Other | Total | Ref |
|---|---|---|---|---|---|---|
| Thailand | 1/195 | 5/449 | 0/4 | 1/7 ‡ | | *Putaporntip et al., 2010* |
| | 0/21 | • | • | • | | *Seethamchai et al., 2008* |
| | 0/4 | • | • | • | | *Fungfuang et al., 2020* |
| | 0/32 | 0/25* | 1/32 | • | | *Fungfuang et al., 2020* |
| | 0/78 | • | • | • | | *Seethamchai et al., 2008* |
| | 0/649 | • | • | • | 8/1496 | *Kaewchot et al., 2022* |
| Philippines | 18/95 | • | • | • | | *Gamalo et al., 2019* |
| | 0/68 | • | • | • | 18/163 | *Zhang et al., 2016* |
| Singapore | 3/13 | • | • | • | | *Jeslyn et al., 2011* |
| | 145/1039 | • | • | • | | *Li et al., 2021b* |
| | 0/40 | • | • | • | 148/1092 | *Zhang et al., 2016* |
| Laos | 1/44 | • | • | • | 1/44 | *Zhang et al., 2016* |
| Cambodia | 0/54 | • | • | • | 0/54 | *Zhang et al., 2016* |
| Myanmar | 0/45 | • | • | • | 0/45 | *Chang et al., 2011* |
| Total | 743/5720 | 26/557 | 1/36 | 1/9 | 773/6322 | |

*Macaca-leonina (Northern Pig-tailed macaque, recently classified as separate species).
†Presbytis spp.
‡Trachypithecus obscurus (Dusky leaf monkey).

**Appendix 1—table 4.** Characteristics of primates tested and number/percentage of confirmed *P. knowlesi* infections (Pk+).

| | | N | (%)* | Pk+ | Pk+ (%)† | CI95% ‡ |
|---|---|---|---|---|---|---|
| Species | M. fascicularis | 5720 | (90.5%) | 745 | 13.0% | (12.2–13.9) |
| | M. nemestrina | 532 | (8.4%) | 26 | 4.9% | (3.4–7.1) |
| | M. leonina | 25 | (0.4%) | 0 | 0.0% | (0.0–13.3) |
| | M. arctoides | 36 | (0.6%) | 1 | 2.8% | (0.5–14.2) |
| | T. obscurus | 7 | (0.1%) | 1 | 14.3% | (2.6–51.3) |
| | Presbytis spp. | 2 | (0.03%) | 0 | 0.0% | (0.0–65.8) |
| Area | Forest | 1740 | (27.5%) | 253 | 14.5% | (13.0–16.3) |
| | Agriculture | 491 | (7.8%) | 72 | 14.7% | (11.8–18.1) |
| | Peri-domestic | 2192 | (34.7%) | 341 | 15.6% | (14.1–17.1) |
| | Urban | 1143 | (18.1%) | 56 | 4.9% | (3.8–6.3) |
| | Sanctuary | 109 | (1.7%) | 5 | 4.6% | (2.0–10.3) |
| | Unspecified | 647 | (10.2%) | 46 | 7.1% | (5.4–9.4) |
| Status | Wild | 6183 | (97.8%) | 768 | 12.4% | (11.6–13.3) |
| | Captive | 139 | (2.2 %) | 5 | 3.6% | (1.5–8.1) |
| Region | Pen. Malaysia | 3069 | (48.5%) | 473 | 15.4% | (14.2–16.7) |
| | Borneo | 251 | (4.0%) | 119 | 47.4% | (41.3–53.6) |
| | Sumatra | 108 | (1.7%) | 6 | 5.5% | (2.6–11.6) |
| | Thailand | 1496 | (23.7%) | 8 | 0.5% | (0.3–1.1) |

*Appendix 1—table 4 Continued on next page*

*Appendix 1—table 4 Continued*

|  | | N | (%)* | Pk+ | Pk+ (%)† | CI95% ‡ |
|---|---|---|---|---|---|---|
|  | Philippines | 163 | (2.6 %) | 18 | 11.0% | (7.1–16.8) |
|  | Singapore | 1092 | (17.3%) | 148 | 13.6% | (11.7–15.7) |
|  | Cambodia | 54 | (0.9%) | 0 | 0.0% | (0.0–6.6) |
|  | Laos | 44 | (0.7%) | 1 | 2.3% | (0.4–11.8) |
|  | Myanmar | 45 | (0.7%) | 0 | 0.0% | (0.0–7.9) |
| Total | | 6322 | (100%) | 773 | 12.2% | (11.4–13.1) |

*Percentage of total number of primates tested (column %).

†Proportion of N positive for *P. knowlesi* (row %).

‡95% confidence interval (CI95%) calculated in R using count and sample size (binomial distribution).

## Appendix 2

### Quality appraisal

Quality was assessed using the Joanna Briggs Institute (JBI) Critical Appraisal tool for prevalence studies (*Munn et al., 2015*). Studies were assessed on nine standardised criteria used to inform inclusion in the meta-analysis. Full criteria and examples of scoring are given in *Appendix 2—tables 1 and 2*. Studies assessed to be of lower quality were those that omitted key information about the sampling method . Two studies were considered to be of higher quality owing to robust sampling and completeness of evidence (*Lee et al., 2011*; *Saleh Huddin et al., 2019*). Given the objective to assess variation in reported prevalence, and in considering the limited usefulness of criteria designed for human participants, the reliable diagnostic methods identified in all studies and the appreciable limitations in surveying wild animals, data from all studies (n=148 estimates) were included for further analyses.

**Appendix 2—table 1.** JBI criteria for assessing bias in meta-analyses of prevalence studies.

| Criteria | | Yes | No | Unclear | N/A |
|---|---|---|---|---|---|
| Q1 | Was the sample frame appropriate to address the target population? | | | | |
| Q2 | Were study participants sampled in an appropriate way? | | | | |
| Q3 | Was the sample size adequate? | | | | |
| Q4 | Were the study subjects and the setting described in detail? | | | | |
| Q5 | Was the data analysis conducted with sufficient coverage of the identified sample? | | | | |
| Q6 | Were valid methods used for the identification of the condition? | | | | |
| Q7 | Was the condition measured in a standard, reliable way for all participants? | | | | |
| Q8 | Was there appropriate statistical analysis? | | | | |
| Q9 | Was the response rate adequate, and if not, was the low response rate managed appropriately? | | | | |

**Appendix 2—table 2.** Example rationale for quality appraisal.

| Sample question | | Example | Assessment |
|---|---|---|---|
| Q1 | Was the sample frame appropriate to address the target population? | Wild animal | Yes |
| | | Captive animal | No |
| | | Not specified | Uncertain |
| Q2 | Were study participants sampled in an appropriate way? | Trapped for study | Yes |
| | | Routine collection | No |
| | | Not specified | Uncertain |
| Q9 | Was the response rate adequate? | Primate data | N/A |

# Appendix 3

## Meta analysis

Sub-group analysis by region was conducted under a random-effects model. Pooled estimates are then back-transformed for interpretation. The Freeman-Tukey double arcsine method is recommended in the transformation of prevalence (*Munn et al., 2015*). However, recent studies have found that back-transformation of the Freeman-Turkey method can generate misleading results, owing to the requirement for a global sample size for inversion (*Schwarzer et al., 2019*). A sensitivity analysis conducted using the logit transformation and untransformed proportions (*Appendix 3—table 1*) revealed a deficit in the back-transformation of the pooled prevalence estimate for Thailand under the Freeman-Tukey transformation, generating a null point estimate. To avoid this error, meta-analysis was conducted under the logit transformation with the inverse variance estimator to account for individual study weighting.

**Appendix 3—table 1.** Sensitivity analysis for transformation of *P. knowlesi* prevalence estimate under random-effects model, shown overall and for Thailand subgroup analysis.

| Method | Overall (*k*=148) | | Subgroup (Thailand, *k*=21) | |
|---|---|---|---|---|
| | P* | CI95% | P | CI95% |
| Freeman-Turkey double arcsine | 0.0943 | (0.0641–0.1284) | 0.0000 | (0.0000–0.0000) |
| Logit | 0.1199 | (0.0935–0.1526) | 0.0199 | (0.0113–0.0346) |
| Untransformed | 0.1415 | (0.1101–0.1730) | 0.0022 | (0.0000–0.0059) |

*Estimated proportion

Small study effects, including selection and publication bias, were assessed by examining funnel plots and imputing 'missing' estimates using the trim-and-fill method (*Lin and Chu, 2018*). Funnel plots were generated using both SE and study size as metrics of variance, as study size has been shown to be more accurate for meta-analyses of proportions where raw estimates tend towards 0 or 1 (*Munn et al., 2015*). Funnel plots for the disaggregated dataset are shown using SE as the variance estimate in *Appendix 3—figure 1*. Asymmetry is highlighted by the trim-and-fill interpolation method, which provides an estimate of missing data and added an additional 54 imputed points to the plot.

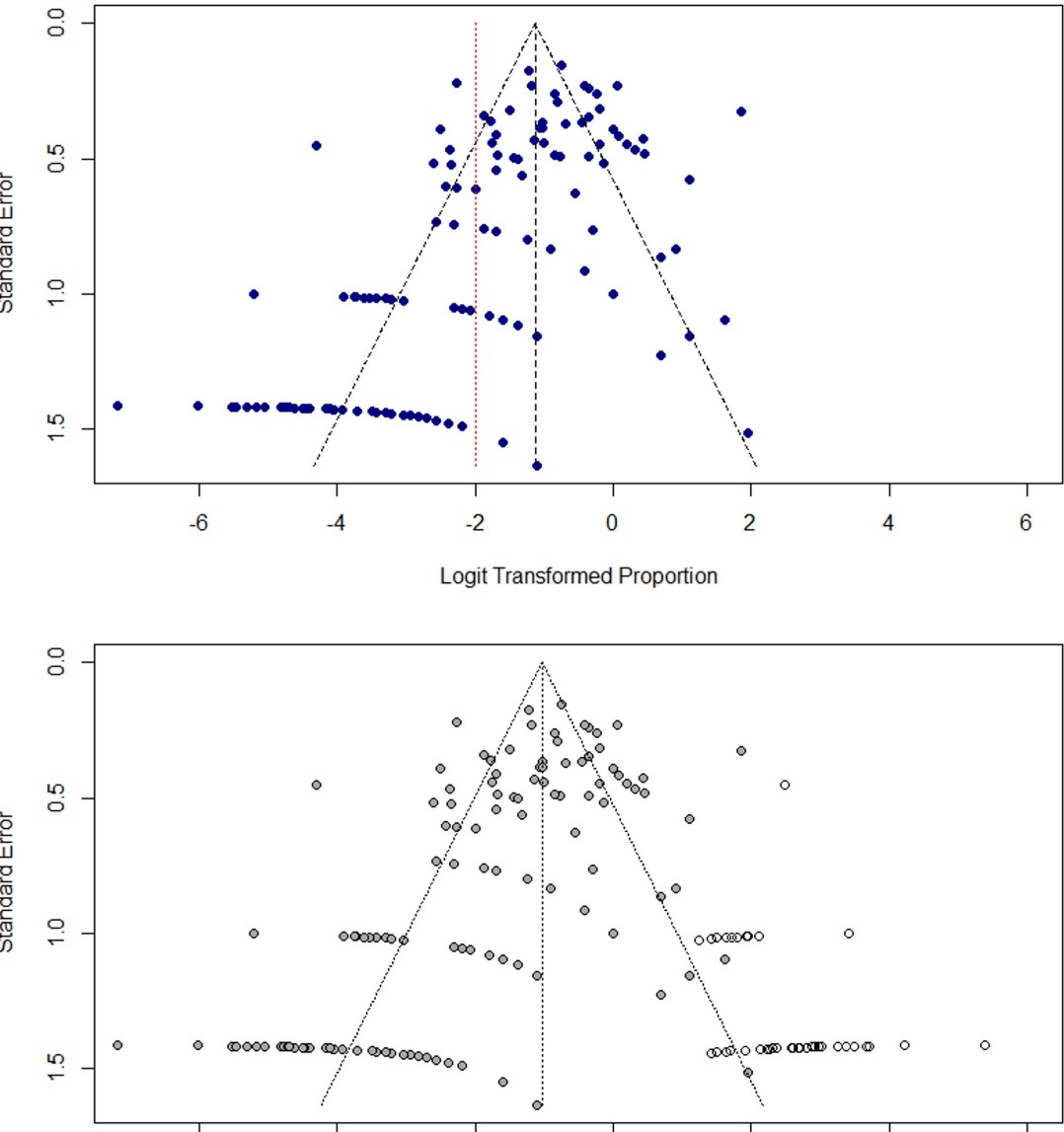

**Appendix 3—figure 1.** Assessment of small study effects in meta-analysis. (**A**) Funnel plot of transformed prevalence (%) against standard error (SE) for study sites (**B**) Funnel plot with imputed data to illustrate asymmetry using trim-and-fill method.

Meta-analysis was conducted with data disaggregated by survey location and primate species (*k*=148). Forest plot of individual study prevalence, presented with pooled regional prevalence estimates and relative sampling effort for the disaggregated data can be visualised in *Appendix 3—figure 2*.

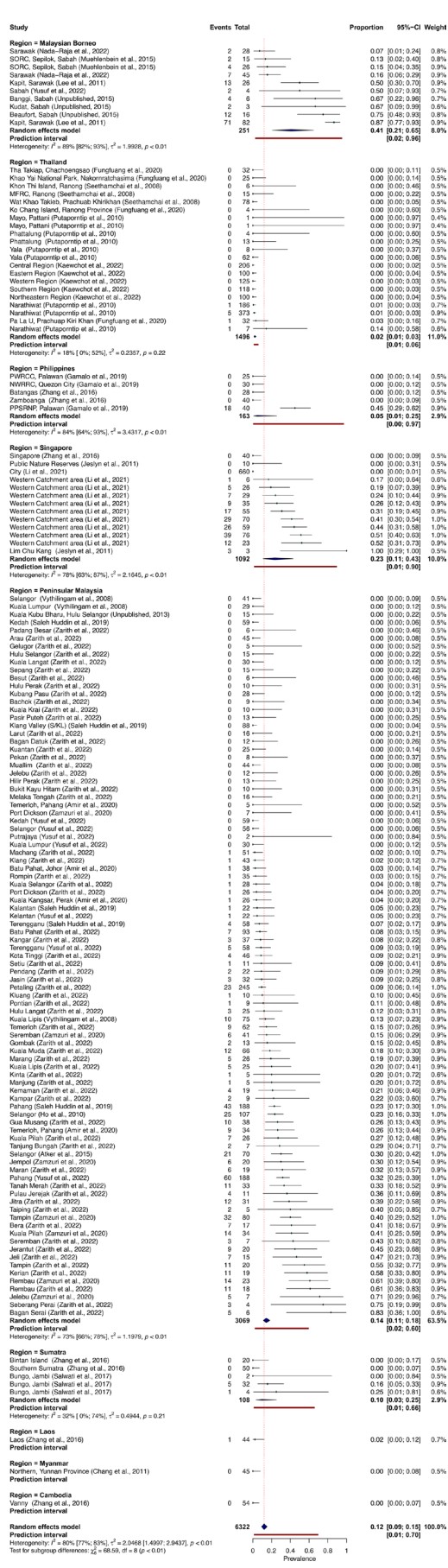

**Appendix 3—figure 2.** Forest plot of *P. knowlesi* prevalence (%) in all species of NHPs in Southeast Asia, disaggregated by species and sampling site, including 95% confidence intervals and individual study weighting. Random-effects analysis, sub-grouped by region. N=148. Zarith et al. refers to personal correspondence derived from the reference *Shahar, 2019*.

## Appendix 4

### Remote sensing data and covariate assembly

Environmental covariates were extracted from satellite-derived remote-sensing datasets. Elevation can be used as a proxy for vector range, with malaria transmission patterns often correspond to altitudinal ranges of vector species (*Hawkes et al., 2019*). Estimates of human population density provide a measure of urbanisation, used to examine risks related to human settlement proximity. Metrics of deforestation including canopy cover (cumulative loss) and degree of fragmentation, which are key determinants of macaque habitat selection (*McGarigal et al., 2021*) and of mosquito vector breeding sites and were derived from Hansen's Global Forest Watch (*Hansen et al., 2013*) Land use classification maps derived from the Intact Forest Landscape (IFL) (*Potapov et al., 2008*) and Copernicus Global Land Service (100 m) (*Buchhorn et al., 2020*) were also explored to provide more detailed information on specific landscape composition. However, as 77.0% of NHP records were collected before the earliest land classification in 2015 (N=114/148), datasets were of limited utility and not pursued further.

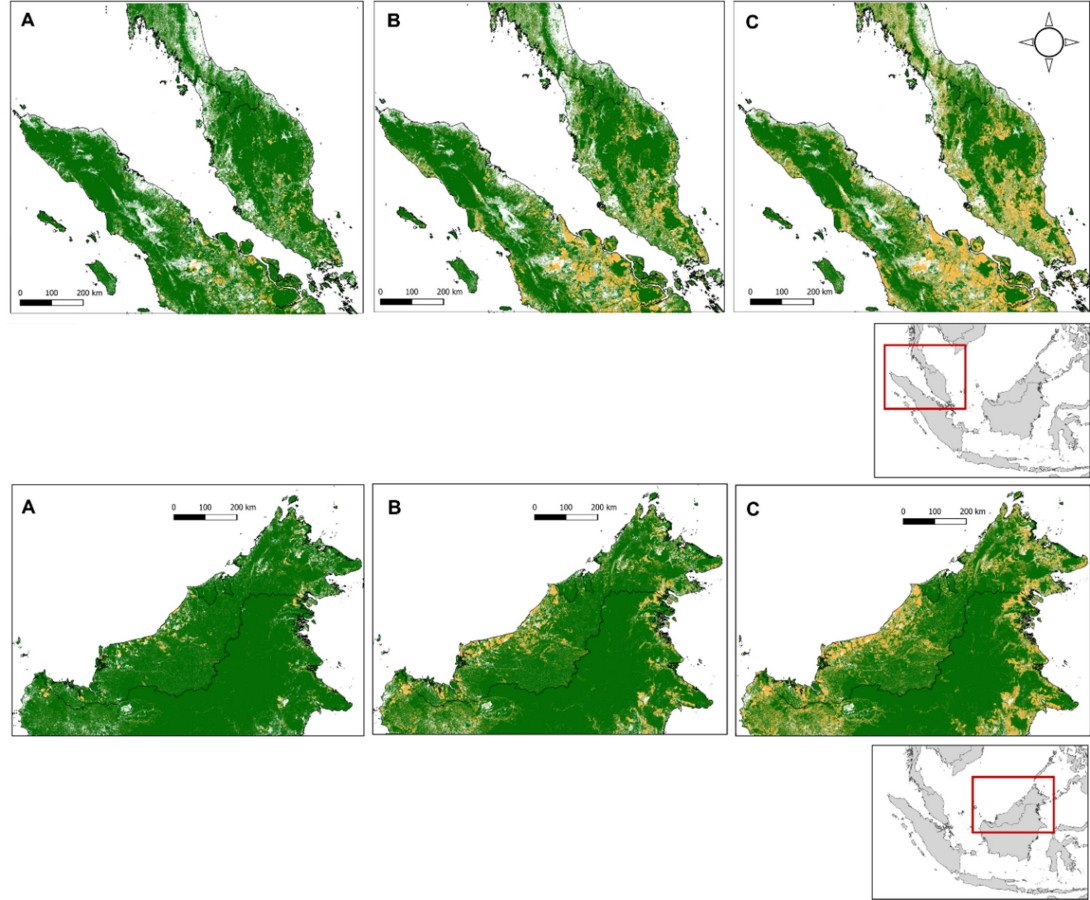

**Appendix 4—figure 1.** Recent forest loss in Peninsular Malaysia (first row) and Malaysian Borneo (secoond row), shown for the years (A) 2006 (B) 2012 and (C) 2019.

Gridded UN-adjusted human population estimates were assembled at 1 km resolution from *WorldPop, 2018* for multiple timepoints as a measure of urbanisation, a proxy for risks related to human settlement proximity. As minimal variation between timepoints was observed, only 2012 (median year of primate data collection) was retained.

**Appendix 4—table 1.** Environmental covariates assembled for regression analysis. Summary values extracted for each covariate within 5, 10, and 20 km circular buffers during processing.

| Covariate | Description | Metric | Resolution | | Processing | Source |
| --- | --- | --- | --- | --- | --- | --- |
| | | | Spatial | Temporal | | |
| Population | UN-adjusted gridded posterior population model estimates at 30 arc-seconds resolution | Person count/1 km$^2$ | 1 km | 2000 2012 2019 | Population density reclassified as high/low (≤300 persons/km$^2$) in QGIS | *WorldPop, 2018* Downloaded as tiff files per country in AOI for years 2000/2012/2019 |
| Elevation | Mean height above sea level | m | 1 km | 2003 | Mean and SD of continuous elevation per radii. Mean-centred and scaled. Categorised into discrete classifications: low (≤200 m), moderate (200–500 m) or high elevation (>500 m) | NASA SRTM 90 m Digital Elevation Database v4.1 (CGIAR-CSI) *Jarvis et al., 2008*. Downloaded as a tiff file at 1 km resampled resolution |
| Forest | Percentage canopy cover per grid cell. Derived from tree cover (vegetation >5 m) and loss (forested to non-forested) | 0–1 | 30 m | Annual 2006–2020 | Tree cover classified as ≥50% crown density per raster cell, generating binary raster (1=forest, 0=non-forest). Annual cover calculated by subtracting cumulative loss per year 2006–2019. Data records matched to reclassified tile by geolocation and year. Posterior proportions categorised as high (>50%) medium (20–50%) or low (≤20%) | Hansen's Global Forest Watch, 30 m resolution Landsat imagery *Hansen et al., 2013*. Tiles downloaded as tiff files for each year 2006–2019 to cover AOI |
| Fragmentation (perimeter: area ratio, PARA) | Perimeter length (m) to patch area (m$^2$) ratio for contiguous forest cover *McGarigal et al., 2021* within buffer | PARA >0 | 30 m | Annual 2006–2020 | Extracted from annual reclassified tree cover rasters within 5, 10 and 20 km circular buffers Output categorised into quartiles | Hansen's Global Forest Watch *Hansen et al., 2013* (as above) |

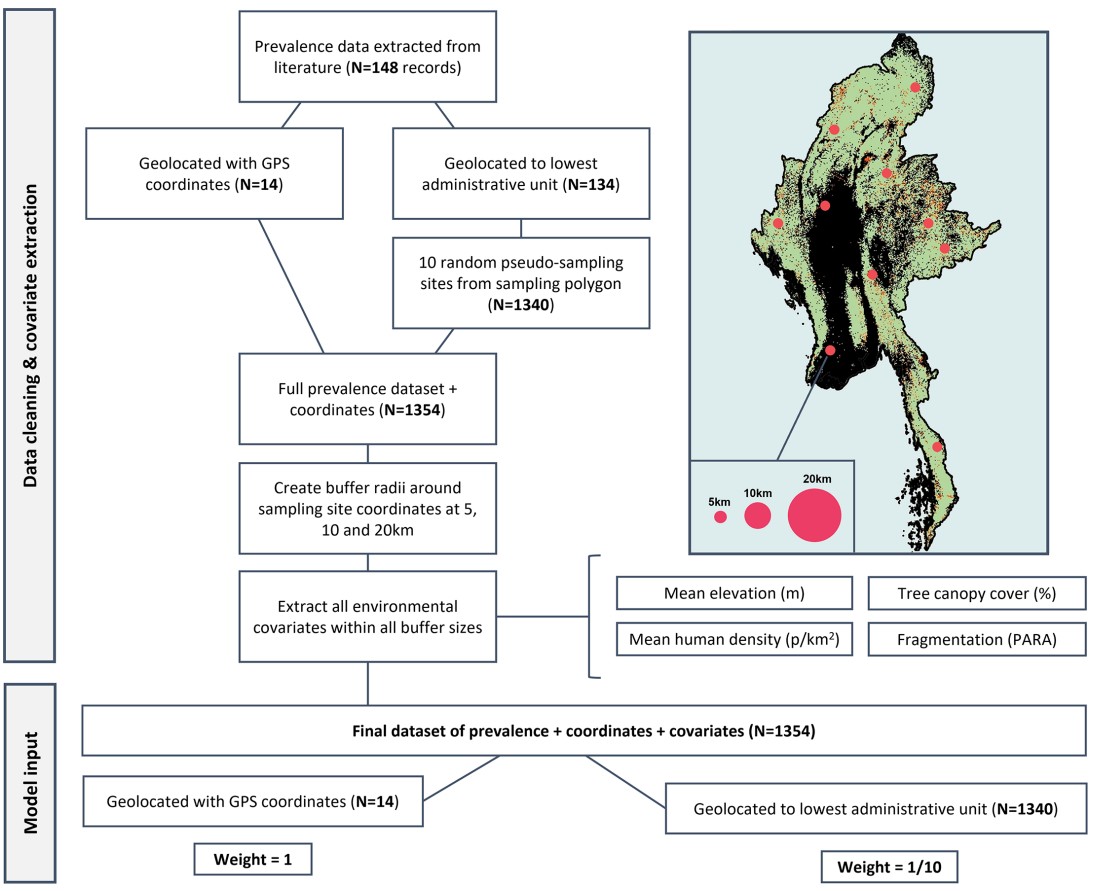

**Appendix 4—figure 2.** Flowchart of data processing. Details of pseudo-sampling and environmental covariate extraction at multiple spatial scales to create final weighted dataset (N=1354).

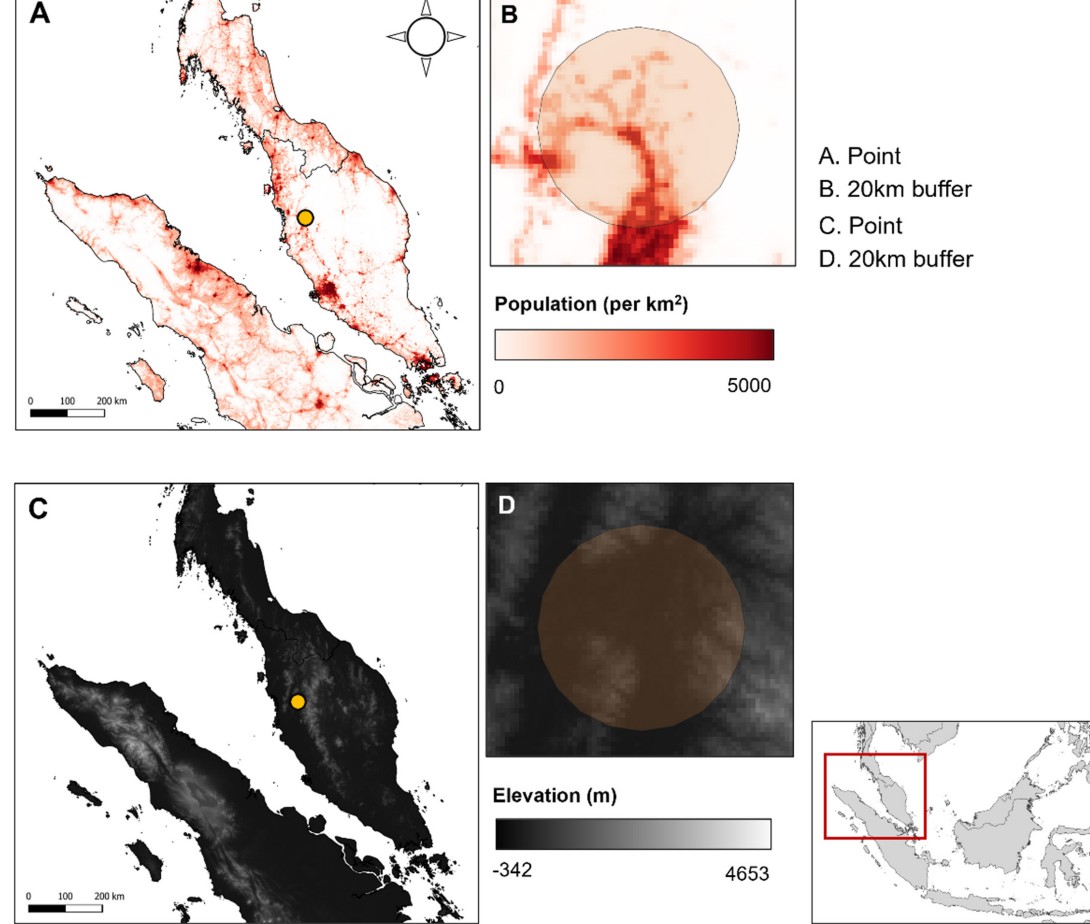

**Appendix 4—figure 3.** Example covariate resolutions in Peninsular Malaysia. (**A**) Data point and (**B**) 20 km buffer over population density layer, 1 km resolution. (**C**) Data point and (**D**) 20 km buffer over SRTM elevation layer, 1 km resolution.

Elevation, population density and forest cover all varied markedly across surveyed sites. Forest cover ranges from negligible to near total cover within 5 km, and up to 99.96% and 99.64% within 10 km and 20 km respectively (*Appendix 4—table 2*, *Appendix 4—figure 4*). Within a 20 km buffer, 46.1% of sites have dense forest cover ≥50% (n=683/1480) and 83.85% have moderate or high forest cover (≥20%), with similar distributions over 5 km and 10 km. Example buffers over forest cover data can be visualised in *Appendix 4—figure 5*.

**Appendix 4—table 2.** Summary of forest cover data (N=1480).

|  | Mean | SD | Range |
| --- | --- | --- | --- |
| Forest cover (5 km) | 50.20% | ±29.29% | 0.00–100.00% |
| Forest cover (10 km) | 49.68% | ±27.30% | 0.00–99.96% |
| Forest cover (20 km) | 48.29% | ±25.35% | 0.00–99.64% |
| Total |  |  | 1480 (100%) |

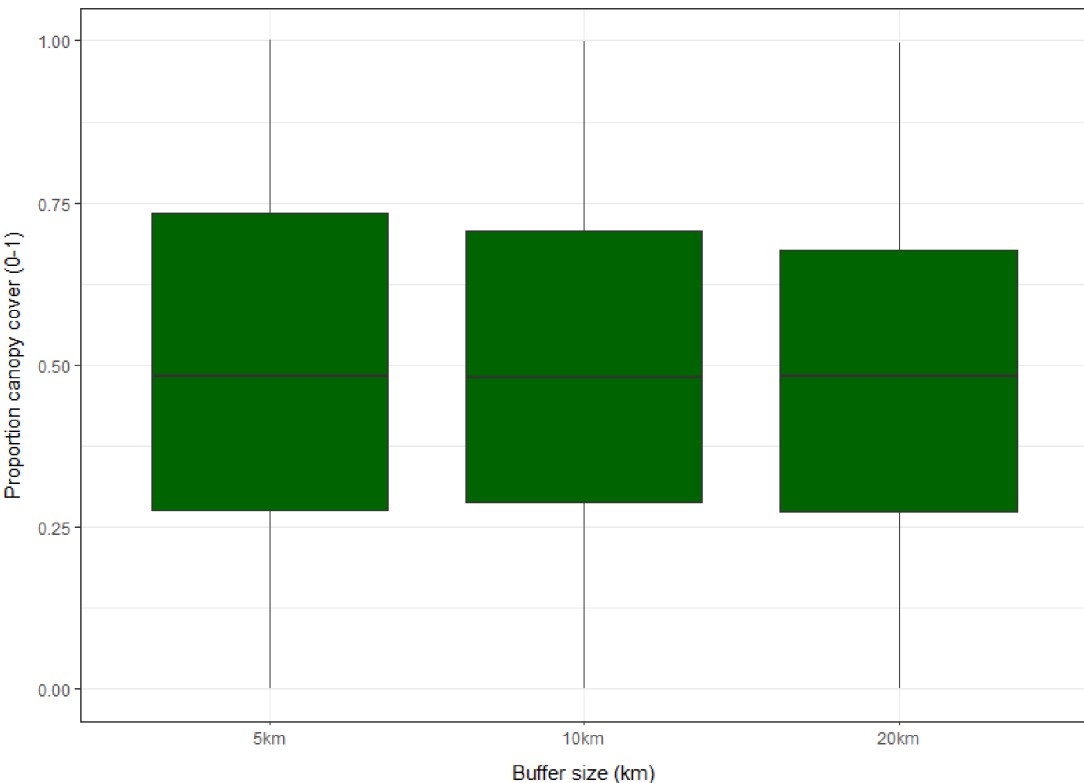

**Appendix 4—figure 4.** Boxplots showing distribution and interquartile range (IQR) of proportional forest cover (0–1) for sampling sites within 5, 10 and 20 km circular buffers across all sites (N=1354).

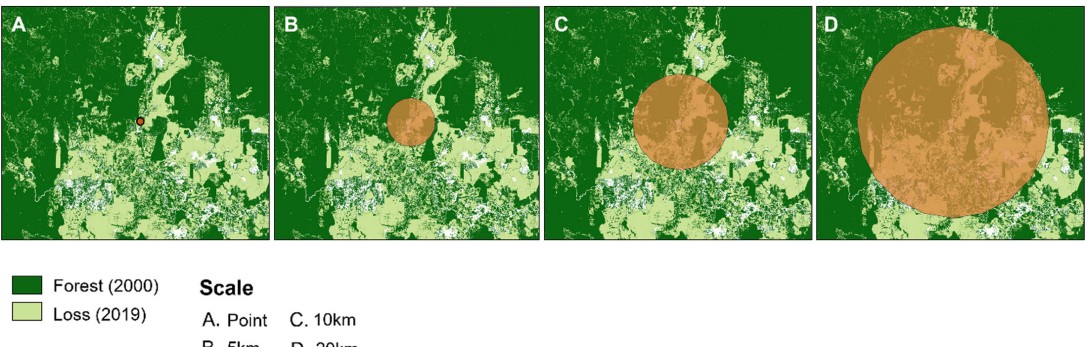

**Appendix 4—figure 5.** Examples of buffer zones around macaque sample sites. Shown over forest cover for 2019 (*Hansen et al., 2013*).

## Appendix 5

### Spatial uncertainty

Available spatial resolution of the survey sites varied. 14 records (9.5%, n=14/148) could be geolocated to a point, using geographic coordinates provided or inferred. The remaining 134 were geolocated to the lowest administrative polygon according to GADM boundary definitions (*Appendix 5—table 1*).

**Appendix 5—table 1.** Geo-positioning of available primate survey data.

|  | Resolution/GADM* | Records/n | Primates/N | Min. (km²)† | Max. (km²) |
|---|---|---|---|---|---|
| Polygon | Country/GID0 | 6 (4.9%) | 853 (17.3%) | 700 | 77,650 |
|  | State/GID1 | 40 (22.0%) | 2699 (32.2%) | 130 | 87,860 |
|  | District/GID2 | 88 (61.8%) | 2433 (43.6%) | 270 | 15,890 |
| Point | GCS ‡ | 14 (11.4%) | 337 (6.8%) | – | – |
| Total |  | 148 (100%) | 4931 (100%) |  |  |

*Administrative boundaries, as classified by GADM (v3.6).
†Minimum and maximum size (km²) of polygons containing *P. knowlesi* data at each admin level.
‡Geographic Coordinate System.

Crude sensitivity analyses were initially conducted to evaluate use of centroids vs random points to approximate macaque survey site. Using GADM classifications, the largest polygon containing NHP data was identified at each administrative level. 10 points were randomly generated within each polygon in QGIS, with buffers at 5/10/20 km. Proportion of forest cover per buffer was extracted, categorised and compared to the forest cover for the centroid (*Appendix 5—figure 1*).

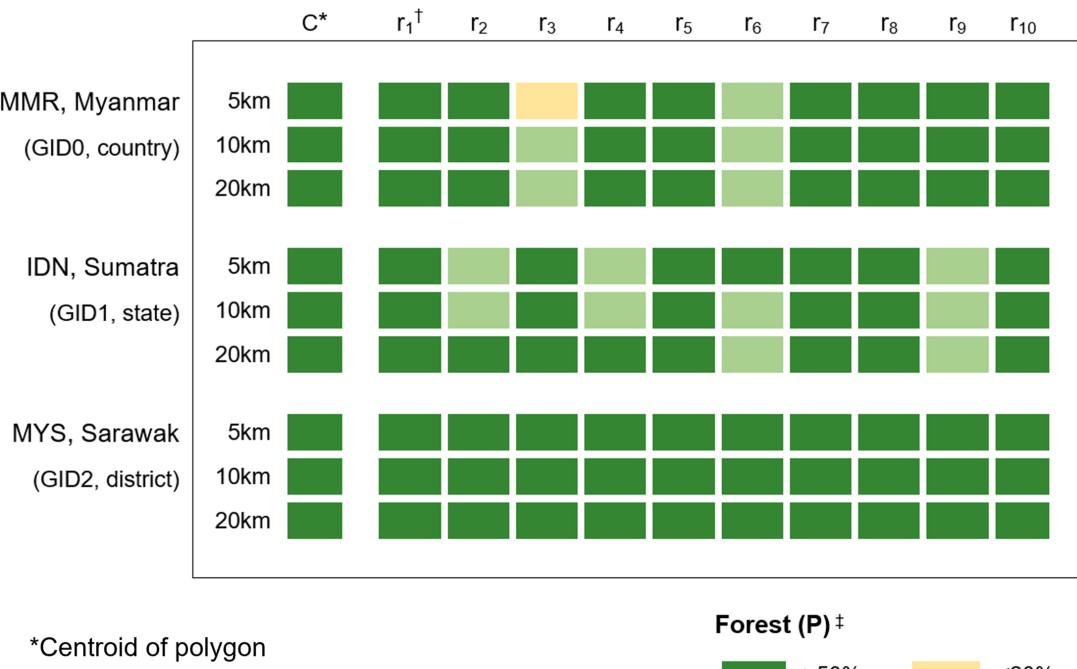

*Centroid of polygon
†Random points (r₁₋₁₀) within polygon
‡Proportion of buffer covered by forest (0<P<1)

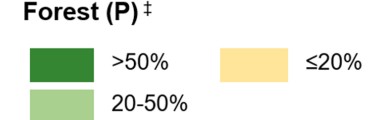

**Appendix 5—figure 1.** Sensitivity analysis comparing centroid forest cover to 10 randomly generated points, shown per radius for the largest polygon at each GADM level.

Results indicate that at district level (GID2), minimal change in forest cover between the points was observable (5 km: 0.68–0.96). However, for both the state of Southern Sumatra (GID1 5 km:

0.26–0.97) and for Myanmar (GID0 5 km: 0.16–0.99) variation was observed, with several points classifying as 'moderate' (20–50% cover) rather than 'high' (>50%) as suggested by the centroid. Overall, results show a disparity between covariates obtained at a central point compared with random points within larger administrative polygons, indicating inadequate sensitivity of centroids as a proxy for local landscape variables where there is spatial uncertainty (*Cheng et al., 2021*).

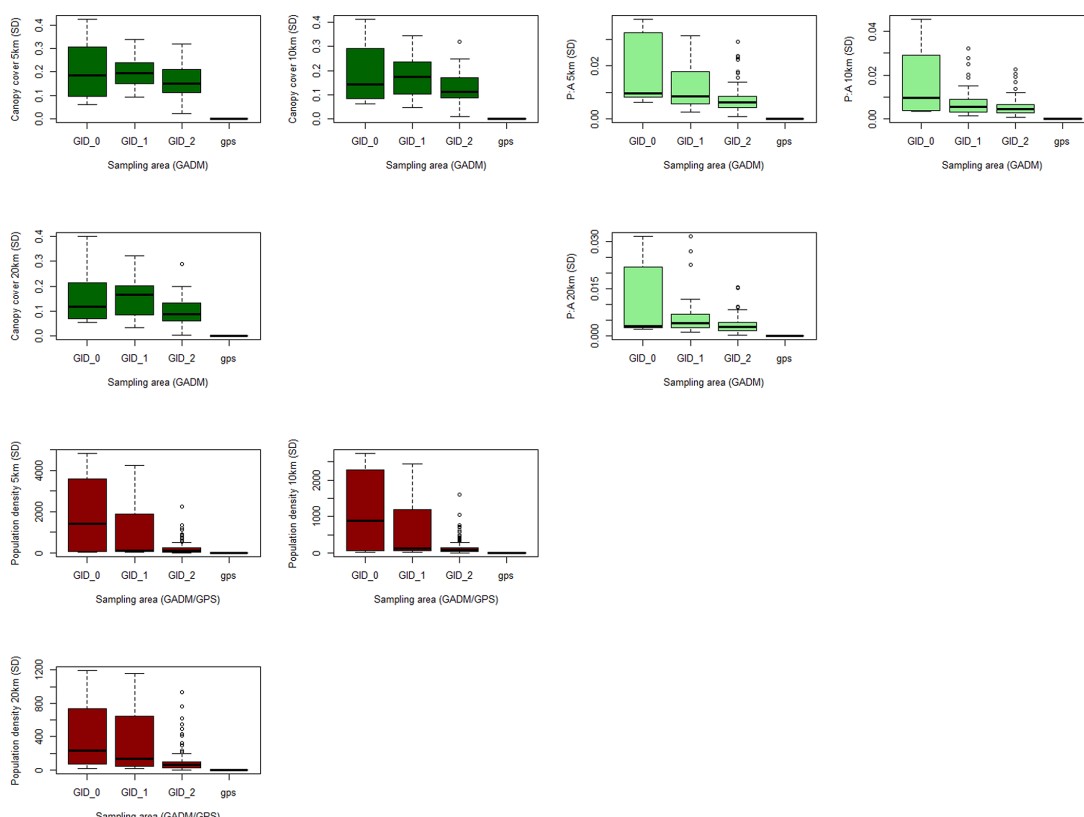

**Appendix 5—figure 2.** Boxplots of standard deviation in repeat sampling of covariates at multiple buffer and boundary sizes. Standard deviation of environmental covariates across 10 sampling site realisations within 5/10/20km buffers, grouped by administrative boundary size or GPS coordinates. Shown for (**A**) canopy cover (%) (**B**) forest fragmentation (P: A ratio) and (**C**) human population density (p/km²).

## Appendix 6

### Regression analysis

**Appendix 6—table 1.** Bivariable analysis for *P. knowlesi* in NHP against all covariates at all spatial scales (N=1354).

| Variable | Crude OR | CI95% | | p value[†] |
|---|---|---|---|---|
| **Bivariable analysis** | | | | |
| Elevation (m) * | | | | |
| ≤5 km | 1.18 | (1.07–1.28) | | 0.000562 |
| ≤10 km | 1.20 | (1.09–1.31) | | 0.0001246 |
| ≤20 km | 1.22 | (1.11–1.33) | | 7.23E-05 |
| Human density (p/km$^2$) * | | | | |
| ≤5 km | 0.84 | (0.77–0.92) | | 4.72E-05 |
| ≤10 km | 0.75 | (0.68–0.82) | | 2.70E-12 |
| ≤20 km | 0.71 | (0.63–0.79) | | 1.37E-12 |
| Forest cover (%) * | | | | |
| ≤5 km | 1.34 | (1.21–1.49) | | 1.86E-08 |
| ≤10 km | 1.41 | (1.26–1.57) | | 6.51E-10 |
| ≤20 km | 1.47 | (1.30–1.67) | | 8.66E-10 |
| Fragmentation (PARA) * | | | | |
| ≤5 km | 0.85 | (0.76–0.95) | | 0.003944 |
| ≤10 km | 0.69 | (0.60–0.80) | | 1.80E-07 |
| ≤20 km | 0.67 | (0.57–0.79) | | 5.14E-07 |
| PARA$^2$ * | | | Quadratic term | |
| ≤5 km | 0.69 | (0.60–0.80) | 0.10 | 2.06E-06 |
| ≤10 km | 0.64 | (0.55–0.74) | 0.08 | 2.65E-08 |
| ≤20 km | 0.67 | (0.57–0.78) | 0.03 | 2.78E-06 |
| Host species | | | | |
| Other | Ref | | | |
| *M. fascicularis* | 2.37 | (1.25–4.60) | | 0.007971 |

*Continuous variable, mean-centred and scaled.
[†]p value derived from Likelihood ratio test (LRT).

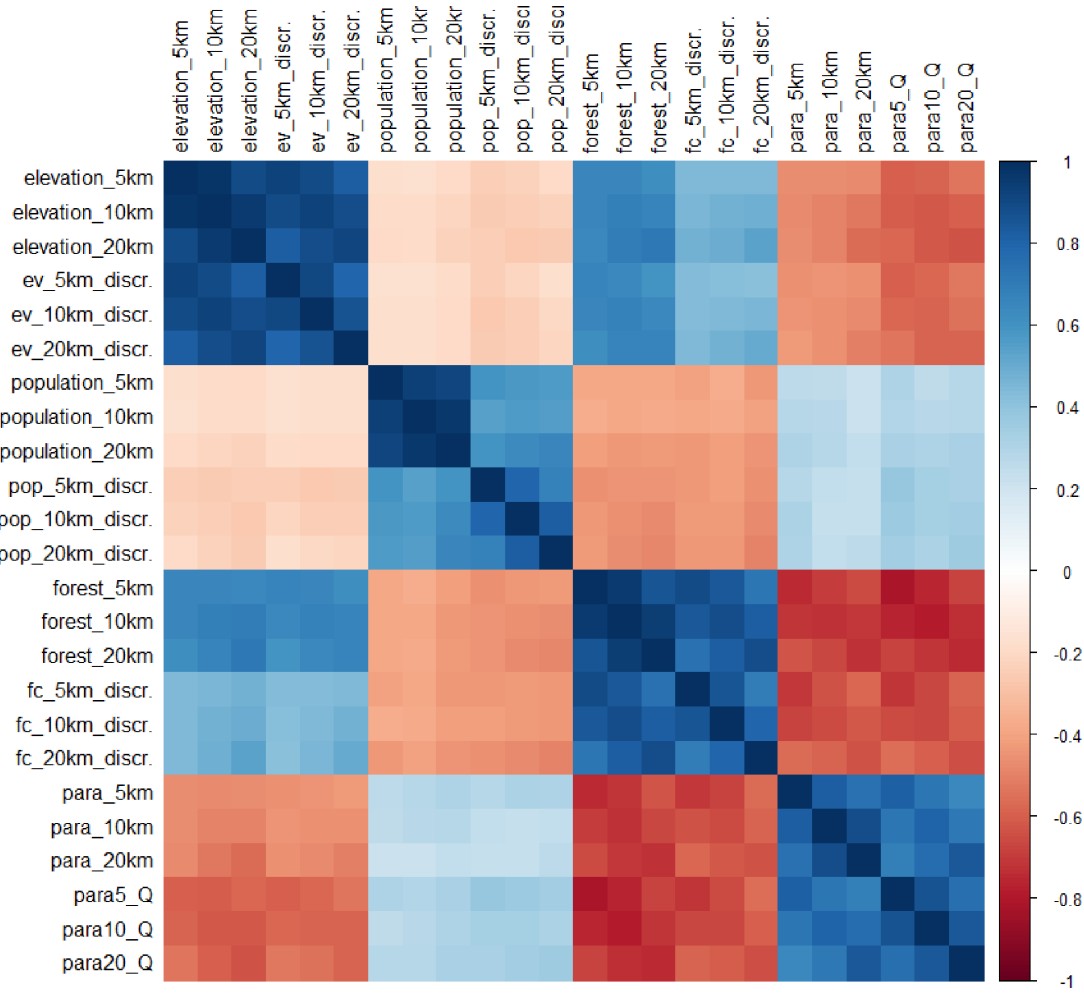

**Appendix 6—figure 1.** Spearman's correlation matrix for all candidate covariates at all spatial scales (n=1354).

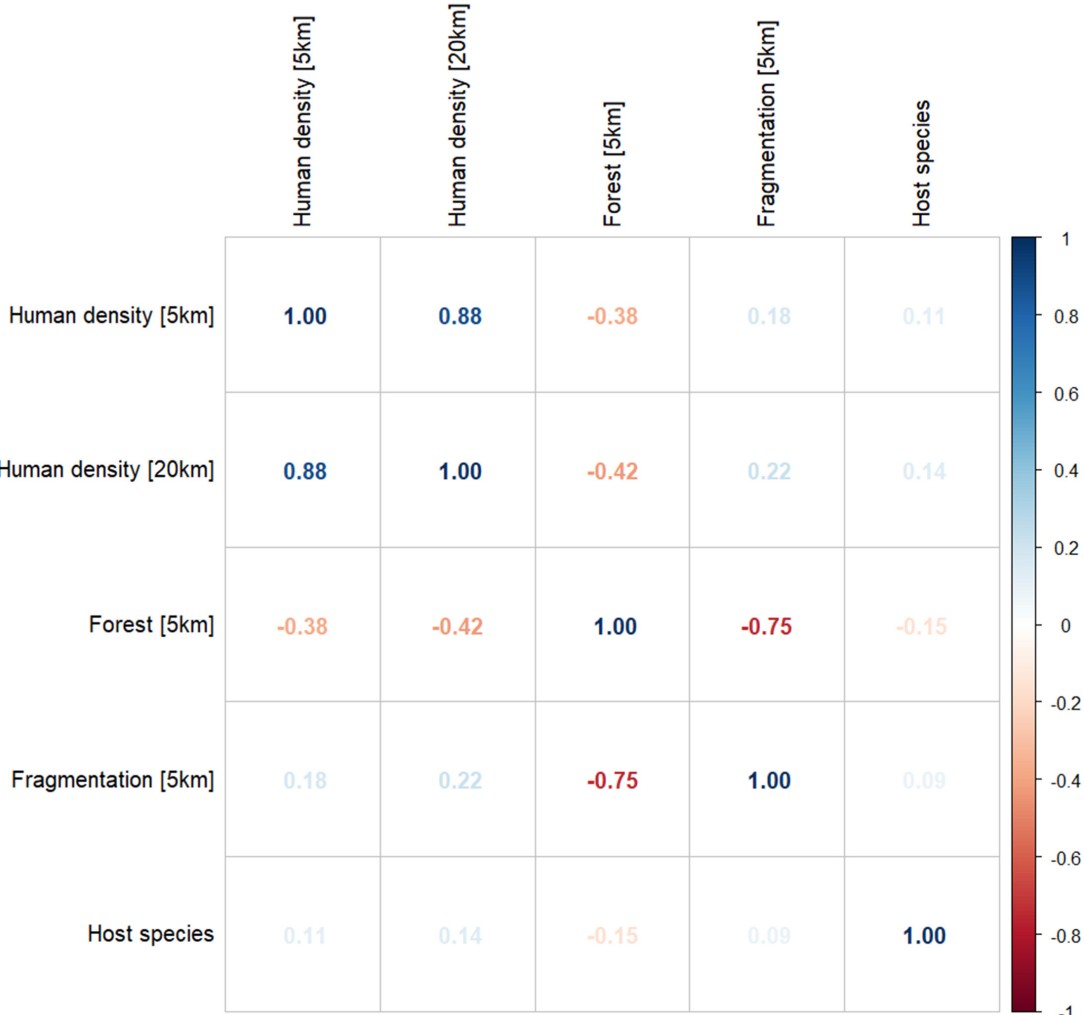

**Appendix 6—figure 2.** Spearman's correlation matrix for covariates at selected spatial scales for final model inclusion (n=1354). Percentage forest cover (5 km) and forest fragmentation (PARA, 5 km) show strong negative correlation ($\rho$=–0.75).

**Appendix 6—table 2.** Multivariable binomial regression analysis of *P. knowlesi* prevalence in NHP with environmental covariates at influential spatial scales, full dataset (N=1354).
AIC = 1229.8.

| Variable | Radius | Multivariable analysis | | |
| | | aOR * | CI95% | p value † |
|---|---|---|---|---|
| Human density (p/km²) ‡ | | | | |
| | ≤5km | 1.36 | (1.16–1.58) | 1.082E-04 |
| | ≤20 km | 0.56 | (0.46–0.67) | 1.311E-10 |
| Forest cover (%) ‡ | | | | |
| | ≤5 km | 1.38 | (1.19–1.60) | 2.046E-05 |
| Fragmentation (PARA) ‡ | | | | |
| | ≤5 km | 1.17 | (1.02–1.34) | 0.0281 |

*Appendix 6—table 2 Continued on next page*

*Appendix 6—table 2 Continued*

| | | Multivariable analysis | | |
|---|---|---|---|---|
| Host species | Other | Ref | | |
| | *M. fascicularis* | 2.50 | (1.31–4.85) | 0.005121 |

*Odds Ratios adjusted for all other variables in the table (aOR). Radius calculated as distance from sample point.
†p value derived from Likelihood ratio test (LRT).
‡Continuous variable, mean-centred and scaled. OR shown per 1 SD increase.

For observations with high geographic uncertainty, random pseudo-sampling of 10 sites (as described in the manuscript and *Appendix 4—figure 2*) was used to avoid overgeneralisations and biases typical when centroids are used as proxy sampling sites[35], with data weighted accordingly. However, this has the potential to introduce extreme or unrealistic values by generating points in landscapes that are outside reasonable estimations of primate study site. Given this, further sensitivity analyses were conducted to validate the results against geographic precision of observations. Data were first truncated to include only data geolocated to administrative boundaries for relatively small area size (see *Appendix 6—table 3*) and exclude highly variable data from country-level boundaries (GID0). Singapore was retained as small administrative unit. GLMM regression models were fit to the truncated dataset.

**Appendix 6—table 3.** Admin boundary sensitivity analysis.
Binomial regression analysis of *P. knowlesi* prevalence in NHP for datapoints assigned to GPS or small sized administrative boundaries (excluding country data) (N=1324).

| AIC = 1221.9 | **Multivariable analysis** | | |
|---|---|---|---|
| | **aOR** | **CI 95%** | **p value (Wald test)*** |
| Human density [5 km] | 1.36 | (1.16–1.58) | *** |
| Human density [20 km] | 0.56 | (0.46–0.67) | *** |
| Forest cover (%) [5 km] | 1.38 | (1.19–1.60) | *** |
| Fragmentation (PARA) [5 km] | 1.18 | (1.02–1.34) | * |
| Host group Other | REF | | |
| *M. fascicularis* | 2.51 | (1.–.31–4.85) | ** |

*Signif. codes: 0 '***' 0.001 '**' 0.01 '*' 0.05 '.' 0.1 ' ' 1.

Administrative boundaries are arbitrary categories and vary considerably in size and landscape consistency. To better evaluate environmental uncertainty associated with each observation, standard deviation (SD) of the covariate values within each set of 10 environmental realisations was calculated (resulting in a single standard deviation value for each covariate at each scale for every prevalence data point). Studies for which the uncertainty (SD) of covariates exceeded half of the maximum standard deviation were censored to avoid spurious associations derived from unreliable/ extreme values for both forest cover (5 km) and fragmentation (5 km). Regression models were fit to the winsorized dataset and compared to results from the full dataset to ensure that associations are robust.

**Appendix 6—table 4.** Distribution of standard deviations across 10 environmental covariates per prevalence data point for landscape variables at all spatial scales (N=1354).

| Covariate | Mean | Range | Median | IQR |
|---|---|---|---|---|
| Canopy [5 km]* | 0.1588 | 0.0000–0.4237 | 0.1534 | 0.1039–0.2277 |
| Canopy [10 km] | 0.1319 | 0.0000–0.4124 | 0.1177 | 0.0827–0.1891 |
| Canopy [20 km] | 0.1051 | 0.0000–0.3999 | 0.0921 | 0.0541–0.1635 |
| Fragmentation [5 km]* | 0.0083 | 0.0000–0.0375 | 0.0063 | 0.0041–0.0094 |
| Fragmentation [10 km] | 0.0061 | 0.0000–0.0455 | 0.0043 | 0.0025–0.0071 |

*Appendix 6—table 4 Continued on next page*

Appendix 6—table 4 Continued

| Covariate | Mean | Range | Median | IQR |
|---|---|---|---|---|
| Fragmentation [20 km] | 0.0041 | 0.0000–0.0316 | 0.0027 | 0.0017–0.0047 |

*Spatial scales selected in final variables.

**Appendix 6—table 5.** Tree canopy cover sensitivity analysis.
Binomial regression of *P. knowlesi* prevalence in NHP for datapoints, with data where SD < ½ the maximum for tree canopy within 5 km (N=814).

| | Multivariable analysis | | |
|---|---|---|---|
| **AIC = 771.1** | **aOR** | **CI 95%** | **p value (Wald test)*** |
| Human density [5 km] | 0.90 | (0.67–1.20) | - |
| Human density [20 km] | 0.72 | (0.50–1.01) | . |
| Forest cover (%) [5 km] | 1.70 | (1.30–2.24) | *** |
| Fragmentation (PARA) [5 km] | 1.38 | (1.01–1.88) | * |
| Host group Other<br>*M. fascicularis* | REF<br>2.63 | (1.35–5.21) | ** |

*Signif. codes: 0 '***' 0.001 '**' 0.01 '*' 0.05 '.' 0.1 '-' 1.

**Appendix 6—table 6.** Landscape fragmentation sensitivity analysis.
Binomial regression of *P. knowlesi* prevalence in NHP for datapoints, with datapoints where SD < ½ the maximum for fragmentation within 5 km (N=1134).

| | Multivariable analysis | | |
|---|---|---|---|
| **AIC = 982.9** | **aOR** | **CI 95%** | **p value (Wald test)*** |
| Human density [5 km] | 0.91 | (0.69–1.18) | - |
| Human density [20 km] | 0.69 | (0.52–0.93) | * |
| Forest cover (%) [5 km] | 1.31 | (1.08–1.60) | ** |
| Fragmentation (PARA) [5 km] | 1.18 | (0.90–1.54) | - |
| Host group Other<br>*M. fascicularis* | REF<br>2.52 | (1.33–4.87) | ** |

*Signif. codes: 0 '***' 0.001 '**' 0.01 '*' 0.05 '.' 0.1 ' ' 1.

Sensitivity analysis has shown that the trends are robust when the data is constrained according to small administrative boundaries or by measures of spatial uncertainty in the environmental variables. However, given that a proportion of points are randomly generated, questions remain about how suitable the resulting sites are for macaque species and consequently whether the associations observed are a realistic indication of ecological trends. To address this, points were subset according to macaque species habitat suitability maps, derived from *Moyes et al., 2016*.

Predicted occurrence maps were combined for three species *Macaca fascicularis*, *Macaca nemestrina* and *Macaca leonina* to create a joint macaque extent for Southeast Asia. Binary maps of predicted habitat extent were then generated using thresholds of moderate (predictions of 0.5 and above), high (>0.75) and very high predicted probability of occurrence (>0.9) (*Moyes et al., 2016*; *Appendix 6—figures 3–5*). Datapoints from the main analysis were then overlaid with each map, and any points (±5 km buffer) that occurred outside of predicted habitat extent for macaque species were removed. Regression models were then fit to the reduced datasets to assess whether associations observed are plausible according to macaque ecology (*Appendix 6—tables 7–9*).

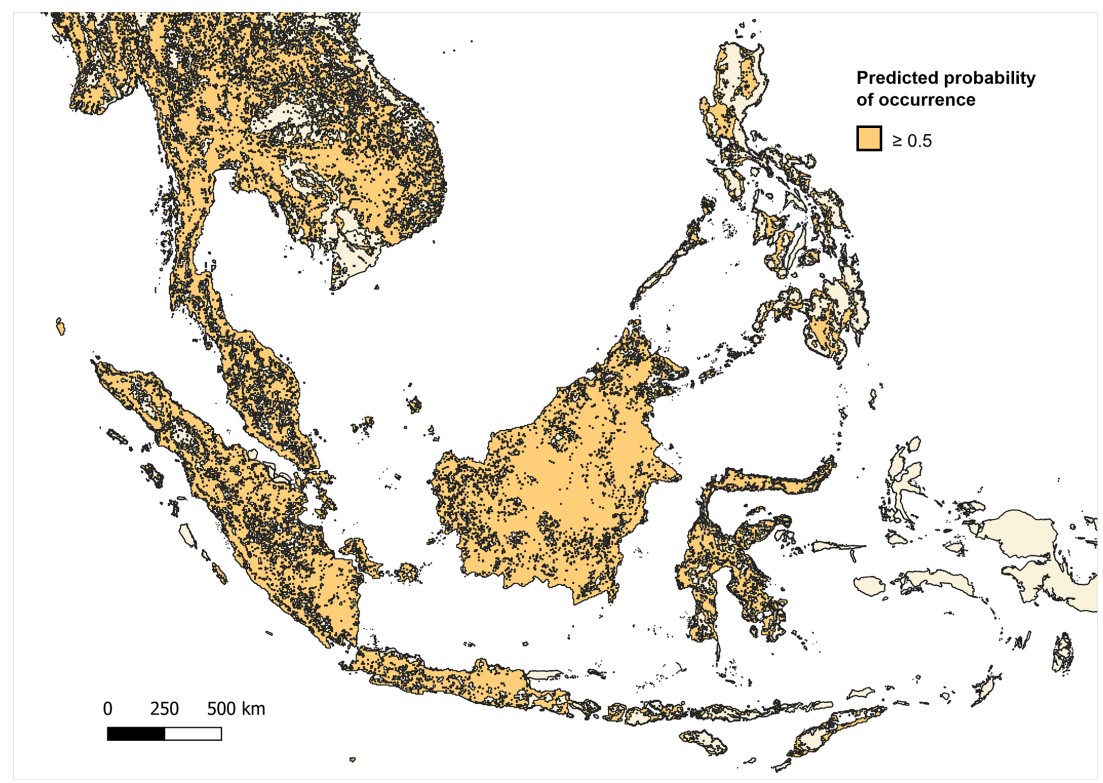

**Appendix 6—figure 3.** Distribution and habitat range of dominant macaque species (M. fascicularis, M. nemestrina, M. leonina) according to predicted probability of occurrence ≥0.5 (on a scale of 0–1.0) per 5x5 km pixel.

**Appendix 6—table 7.** Macaque habitat suitability sensitivity analysis.
Binomial regression of *P. knowlesi* prevalence in NHP for datapoints, including only datapoints with 5 km buffers that intersect with areas with ≥0.5 probability of predicted macaque occurrence (N=1331).

| AIC = 1197.2 | Multivariable analysis | | |
|---|---|---|---|
| | aOR | CI 95% | p value (Wald test)* |
| Human density [5 km] | 1.32 | (1.13–1.54) | *** |
| Human density [20 km] | 0.55 | (0.45–0.66) | *** |
| Forest cover (%) [5 km] | 1.30 | (1.12–1.52) | *** |
| Fragmentation (PARA) [5 km] | 1.12 | (0.97–1.29) | - |
| Host group Other<br>*M. fascicularis* | REF<br>2.48 | <br>(1.31–4.82) | <br>** |

*Signif. codes: 0 '***' 0.001 '**' 0.01 '*' 0.05 '.' 0.1 ' ' 1.

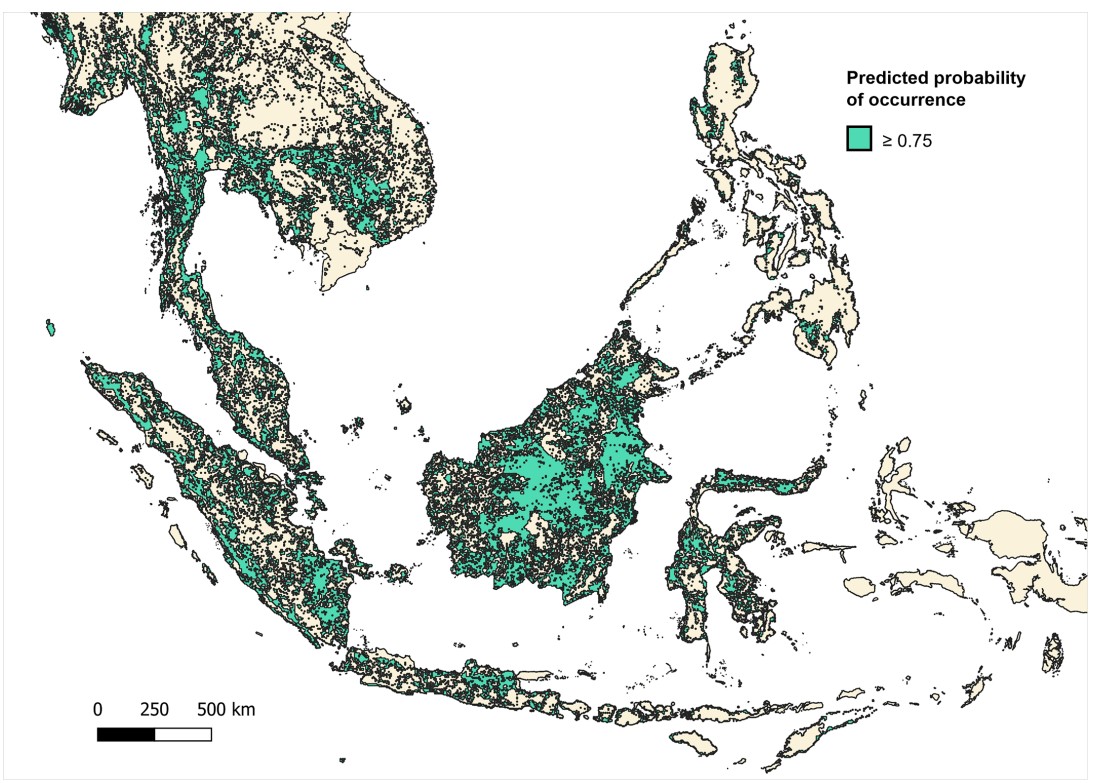

**Appendix 6—figure 4.** Distribution and habitat range of dominant macaque species (M. fascicularis, M. nemestrina, M. leonina) according to predicted probability of occurrence ≥0.75 (on a scale of 0–1.0) per 5x5 km pixel.

**Appendix 6—table 8.** Macaque habitat suitability sensitivity analysis.
Binomial regression of *P. knowlesi* prevalence in NHP for datapoints, including only datapoints with 5 km buffers that intersect with areas with ≥0.75 probability of predicted macaque occurrence (N=1177).

| | Multivariable analysis | | |
|---|---|---|---|
| **AIC = 1115.5** | **aOR** | **CI 95%** | **p value (Wald test)*** |
| Human density [5 km] | 1.34 | (1.14–1.58) | *** |
| Human density [20 km] | 0.57 | (0.47–0.69) | *** |
| Forest cover (%) [5 km] | 1.23 | (1.04–1.47) | * |
| Fragmentation (PARA) [5 km] | 1.04 | (0.86–1.24) | - |
| Host group Other<br>*M. fascicularis* | REF<br>2.69 | (1.38–5.38) | **|

*Signif. codes: 0 '***' 0.001 '**' 0.01 '*' 0.05 '.' 0.1 ' ' 1.

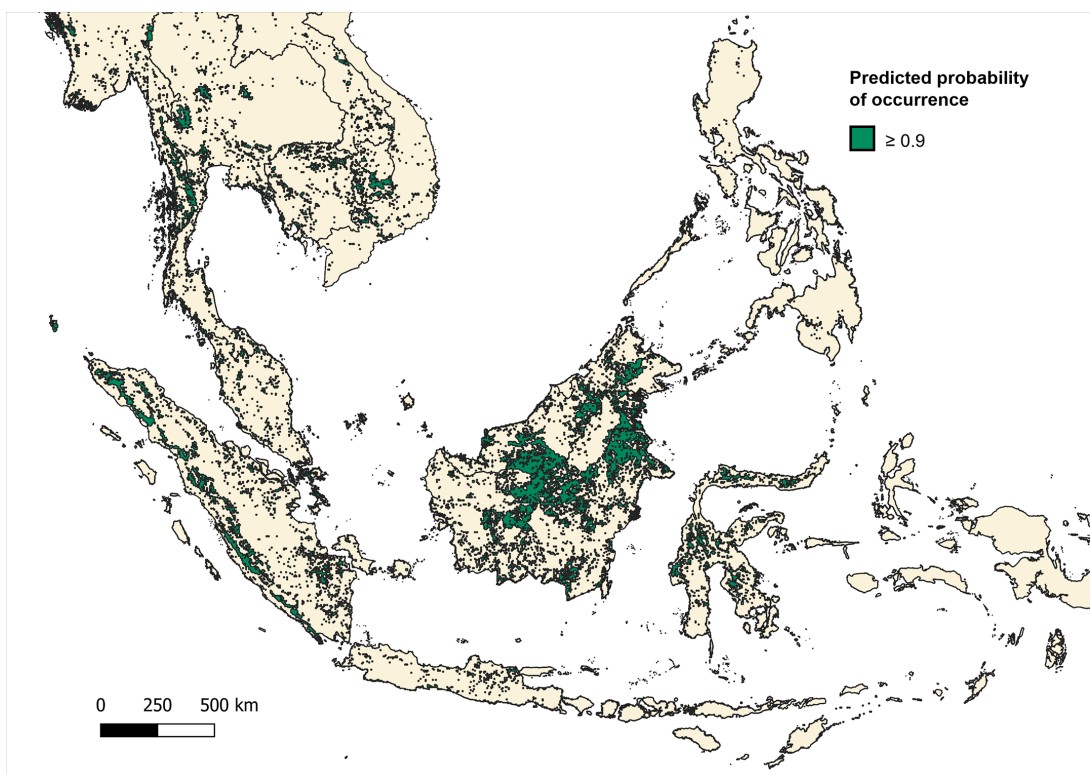

**Appendix 6—figure 5.** Predicted distribution and habitat range of all macaque species (*M. fascicularis*, *M. nemestrina*, *M. leonina*) according to predicted probability of occurrence ≥0.9 (on a scale of 0–1.0) per 5x5 km pixel.

**Appendix 6—table 9.** Macaque habitat suitability sensitivity analysis.
Binomial regression of *P. knowlesi* prevalence in NHP for datapoints, including only datapoints with 5 km buffers that intersect with areas with ≥0.9 probability of predicted macaque occurrence (N=567).

| AIC = 685.2 | Multivariable analysis | | |
| --- | --- | --- | --- |
| | aOR | CI 95% | p value (Wald test)* |
| Human density [5 km] | 1.86 | (1.49–2.32) | *** |
| Human density [20 km] | 0.36 | (0.26–0.49) | *** |
| Forest cover (%) [5 km] | 1.47 | (1.14–1.90) | ** |
| Fragmentation (PARA) [5 km] | 1.35 | (1.02–1.77) | * |
| Host group Other<br>*M. fascicularis* | REF<br>3.13 | (1.50–6.75) | ** |

*Signif. codes: 0 '***' 0.001 '**' 0.01 '*' 0.05 '.' 0.1 ' ' 1.

