## [Editor Report · eLife assessment]

This **useful** study presents findings regarding the impact of forest cover and fragmentation on the prevalence of malaria in non-human primates. The evidence supporting the claims of the authors is **solid**.

---

## [Referee Report · Reviewer #1 (Public Review)]

The paper aims to determine the impact of forest cover and fragmentation on the prevalence of malaria in non-human primates. The paper uses existing spatial datasets, as well as data obtained through published studies on zoonotic malaria. The findings of this study are important, as forest loss is still occurring in the tropics which will impact human infections of zoonotic malaria.

---

## [Referee Report · Reviewer #2 (Public Review)]

This is the first comprehensive study aimed at assessing the impact of landscape modification on the prevalence of *P. knowlesi* malaria in non-human primates in Southeast Asia. This is a very important and timely topic both in terms of developing a better understanding of zoonotic disease spillover and the impact of human modification of landscape on disease prevalence.

This study uses the meta-analysis approach to incorporate the existing data sources into a new and completely independent study that answers novel research questions linked to geospatial data analysis. The challenge, however, is that neither the sampling design of previous studies nor their geospatial accuracy are intended for spatially-explicit assessments of landscape impact. On the one hand, the data collection scheme in existing studies was intentionally opportunistic and does not represent a full range of landscape conditions that would allow for inferring the linkages between landscape parameters and P. knowlesi prevalence in NHP across the region as a whole. On the other hand, the absolute majority of existing studies did not have locational precision in reporting results and thus sweeping assumptions about the landscape representation had to be made for the modeling experiment. Finally, the landscape characterization was oversimplified in this study, making it difficult to extract meaningful relationships between the NHP/human intersection on the landscape and the consequences for *P. knowlesi* malaria transmission and prevalence.

Despite study limitations, the authors point to the critical importance of understanding vector dynamics in fragmented forested landscapes as the likely primary driver in enhanced malaria transmission. This is an important conclusion particularly when taken together with the emerging evidence of substantially different mosquito biting behaviors than previously reported across various geographic regions.

Another important component of this study is its recognition and focus on the value of geospatial analysis and the availability of geospatial data for understanding complex human/environment interactions to enable monitoring and forecasting potential for zoonotic disease spillover into human populations. More multi-disciplinary focus on disease modeling is of crucial importance for current and future goals of eliminating existing and preventing novel disease outbreaks.

---

## [Author Response]

The following is the authors’ response to the previous reviews.

**eLife assessment**
This study presents useful findings regarding the impact of forest cover and fragmentation on the prevalence of malaria in non-human primates. The evidence supporting the claims of the authors is, however, incomplete, as the sampling design cannot adequately address the geospatial issues that this study focuses on.
**Public Reviews:**

**Reviewer #1 (Public Review):**
The study as a concept is well designed, although there is still one issue I see in the methodology.I still have concerns with their attempts to combine the different scales of data. While the use of point data is great, it limits the sample size, and they have included the district to country level data to try and increase the sample size. The problem is that although they try to get an overall estimate at the district/state/country by taking 10 random sample points, which could be a method to get an estimate for the district/state/country. It would be a suitable method if the primates were evenly distributed across the district/state/country. The reality is that the primates are not evenly distributed across the district/state/country therefore the random point sampling is not a reasonable method to get an estimate of the environmental variables in relation to the macaques. For example if you had a mountainous country and you took 10 random points to estimate altitude, you would end up with a large number, but if all the animals of interest lived on the coast, your average altitude is meaningless in relation to the animals of interest as they are all living at low altitude. The fact that the model relies less on highly variable components and places more reliance on less variable components, is really not relevant as the district/state/country measurements have no real meaning in relation to the distribution of masques.A simple possible way forward could be to run the model without the district/state/country samples and see what the outcome is. If the outcome is similar then the random point method may be viable (but if it gives the same outcome as ignoring those samples then you don't need the district/state/country samples). If you get a totally different outcome then it should raise concerns about using the district/state/country samples.This paper is a really nice piece of work and is a valuable contribution but the district/state/country sample issue really needs to be addressed.
**Recommendations for the authors:**

**Reviewer #1 (Recommendations For The Authors):**
A simple possible way forward could be to run the model without the district/state/country samples and see what the outcome is. If the outcome is similar then the random point method may be viable (but if it gives the same outcome as ignoring those samples then you don't need thedistrict/state/country samples). If you get a totally different outcome then it should raise concerns about using the district/state/country samples.

Thank you for your comments, and for the suggestions to address the issues identified in your main commentary by running an analysis on exclusively GPS geolocated data points. This was the original plan for analysis, but the available data identified in the literature review includes only 14 data points (macaque *P. knowlesi* prevalence surveys) with associated GPS coordinates. This was found to be too limited to obtain meaningful results from a regression analysis, and hence we then explored methods for utilising all available data to identify trends whilst accounting for spatial uncertainty in the analysis. As the point location only represents the location of capture and not the extent of the home range of the NHPs, we additionally feel there is value in exploring methods to encompass the wider surrounding habitat.

We do appreciate the concerns you raise with the random point method being used to represent macaque survey sites when species of interest are not necessarily evenly distributed across an area. To investigate this, we ran sensitivity analysis on a subset of the dataset according to whether the points fall in areas of >50%, >75% or >90% predicted probability of macaque occurrence, with maps derived from published models of macaque suitability in Southeast Asia. For each of these thresholds, points that fall outside these areas were removed – such that, if a random point is located on a mountain range where there is 0 likelihood of macaque occurrence, it is excluded from the analysis. We found that restricting analysis to areas with highly probably macaque habitat still shows a robust effect of forest cover on NHP prevalence, and additionally that for the most conservative (>90%) habitat threshold there remains an effect of forest fragmentation on prevalence (Appendix 6—table 9, Appendix 6—figure 5). Given that using the full data set increases the uncertainty, as there is more variation in covariates between the replicates, this can be considered a more conservative approach to detecting an effect of environment as reported in the main findings.